# Ultralight crystalline hybrid composite material for highly efficient sequestration of radioiodine

Sahel Fajal [1], Writakshi Mandal [1], Arun Torris [2], Dipanjan Majumder [1], Sumanta Let [1], Arunabha Sen [1], Fayis Kanheerampockil [2], Mandar M. Shirolkar [3,4] & Sujit K. Ghosh [1,5] ✉

Considering the importance of sustainable nuclear energy, effective management of radioactive nuclear waste, such as sequestration of radioiodine has inflicted a significant research attention in recent years. Despite the fact that materials have been reported for the adsorption of iodine, development of effective adsorbent with significantly improved segregation properties for widespread practical applications still remain exceedingly difficult due to lack of proper design strategies. Herein, utilizing unique hybridization synthetic strategy, a composite crystalline aerogel material has been fabricated by covalent stepping of an amino-functionalized stable cationic discrete metal-organic polyhedra with dual-pore containing imine-functionalized covalent organic framework. The ultralight hybrid composite exhibits large surface area with hierarchical macro-micro porosity and multi-functional binding sites, which collectively interact with iodine. The developed nano-adsorbent demonstrate ultrahigh vapor and aqueous-phase iodine adsorption capacities of 9.98 g.g$^{-1}$ and 4.74 g.g$^{-1}$, respectively, in static conditions with fast adsorption kinetics, high retention efficiency, reusability and recovery.

The limitations associated with the spent nuclear fuel reprocessing and improper disposal of radionuclides wastes have thwarted the further sustainable development of nuclear energy[1–5]. In nuclear plant, during reprocessing and in subsequent steps of spent nuclear fuel by dissolving in hot nitric acid solution, various highly toxic radionuclides are generated[6,7]. Among them, volatile radioactive iodine isotopes (e.g., $^{129}$I and $^{131}$I) has attracted significant attention from the environment and safety point of view, owing to its hazardous impact with radiological (longest half-life $t_{1/2} = 1.6 \times 10^7$ years), chemical (high-mobility) and biological (rapid bio-accumulation) toxicity[8–10]. Moreover, water discharged from nuclear reactor chiller plants contain a significant amount of radioactive iodine, directly contaminating substantial

watery environments[11–14]. Therefore, considering both the importance of sustainable future development of nuclear energy and water purification along with the practical relevance for essential medical uses, efficient and selective sequestration of radioactive iodine both from vapor and aqueous phase counts as a priority research topic[15–17].

To address this cardinal issue, recently, utilizing rational synthetic strategies, pioneering endeavors have been devoted to develop various excellent porous materials for the exceptional enrichment of iodine from contaminated samples[17–31]. Nevertheless, most of the studies are still limited to the exploration of fast kinetics, high capacity, capture at high temperature and improved selectivity, which has actuated researchers to develop novel materials for the efficient

[1]Department of Chemistry, Indian Institute of Science Education and Research (IISER) Pune, Dr. Homi Bhaba Road, Pashan, 411008 Pune, India. [2]Polymer Science and Engineering Division, CSIR-National Chemical Laboratory, Dr. Homi Bhabha Road, Pune 411008, India. [3]Advanced Bio-Agro Tech Pvt. Ltd, Baner, Pune 411045, India. [4]Norel Nutrient Bio-Agro Tech Pvt. Ltd, Baner, Pune 411045, India. [5]Centre for Water Research (CWR), Indian Institute of Science Education and Research (IISER) Pune, Dr. Homi Bhaba Road, Pashan, Pune 411008, India. ✉e-mail: sghosh@iiserpune.ac.in

capture of iodine. Not but what, few of these performance criteria were full-filled by individual adsorbents, due to the scarcity of systematic synthetic strategies, the development of a single material that enables highly selective entrapment of iodine with fast kinetics, high capacity, large retention efficiency, recyclability, and capture in different relevant temperatures from both air and water medium remain an unexplored territory.

To accomplish this, it is important to gain insights into the underlying chemistry of the active interaction sites. This can be revamped by taking into account the specific interactions between iodine and the sorbent material at the molecular level. With this aim, of designing an effective adsorbent for high-performance iodine capture, we reconsidered the previous studies and investigated their insight mechanism towards efficient iodine uptake. The high adsorption capacity of iodine is predominantly governed by the textural features (such as surface area, pore size, and pore volume) of the porous sorbents[26–28]. Moreover, the strong interactions between the electron-deficient iodine molecules with the optimized dense binding sites of electronically rich low-density porous materials are favorable for high iodine uptake, owing to the formation of a suitable charge-transfer complex[23–29]. On the other hand, several individual or collective effects of multiple functionalities, such as induction of pi-electron-rich conjugated frameworks[28], presence of heterocyclic moieties[29], and incorporation of specific heteroatoms[32], have shown clear benefits towards improved $I_2$ capture efficiency. Among the heterocyclic sites, various N-doped structural moieties, including imine, triazine, pyridine, amine, etc. have been infused inside aromatic networks to expedite a large iodine enrichment[26,29]. In addition, integration of cationic functionality bearing free anions inside the adsorbent can trigger the $I_2$ vapor adsorption competence by generating strong electrostatic interactions with in-situ formed polyiodide species[5,25,26]. These framework-integrated counter anions facilitate energetically favorable interactions with iodine in vapor phase[5]. Also, these exchangeable free anions aid in the trapping of $I_3^-$ anions in aqueous phase[12]. Other studies demonstrated that in comparison to its powder counterparts, the strategic fabrication of crystalline aerogels enable greatly enhanced $I_2$ capture with rapid kinetics[33,34]. Furthermore, studies suggest that materials featuring Zr-secondary building units (SBU) with hydroxyl groups exhibit selective strong interactions with iodine than that of other interference e.g., $NO_2$, a major off-gas species (Supplementary Fig. 4)[19].

Integrating all these features in a single material to develop an excellent $I_2/I_3^-$ adsorbent is challenging and remains largely unexplored due to shortfall of systematic synthetic preparation. In this regard, the development of advanced hybrid composite porous materials can be beneficial in order to achieve upgraded performance toward target-specific applications. Through carefully orchestrated molecular engineering, several innovative porous materials have recently been acknowledged as viable platforms for numerous activities due to their predesigned adjustable reticular structure[35,36]. That being said, efficacious performance with collective excellence of different functional materials can be achieved through the fabrication of composite materials[37,38]. Towards this aim, in the case of host-guest hybrid material-based ideal adsorbents, the guest should be evenly dispersed throughout the host matrix and adorned with chelating groups that can strongly interact with the host surface. In this sense, metal–organic polyhedra (MOP) and covalent organic framework (COF), could be the perfect choice for guest and host matrix, respectively. MOPs and COFs are two well-known crystalline porous solid materials, constructed from coordination assembly of metal ions/cluster with organic ligands and covalent threading of pure organic molecules, respectively[39–42]. The unique solution-processable nature of MOPs and high chemical stability with long-range order of COFs promote them as promising compatible materials towards the fabrication of composites. Nevertheless, unfortunately, MOPs fall short of their

conceivable potential pertaining to their inevitable aggregation-induced blocking of the active sites in solid-state after the removal of guest[43]. With this interest, only a few efforts have been performed to alleviate these key issues by encapsulating MOPs into various porous matrix including MOFs and silica or connecting with other polymer membranes[44–46]. However, the development of hybrid composite porous material by covalently strapping of MOPs with COFs, utilizing a unique hybridization strategy and further construction of crystalline hybrid aerogel is still an unfinished challenge.

Taking all these factors into consideration, herein, we designed and fabricated a crystalline hybrid ionic aerogel material (IPcomp-7, where "IP" stands for IISER Pune, and "comp" stands for composite) by covalent grafting of amino functionalized Zr(IV)-MOP with COF, which exhibited highly efficient sequestration efficacy towards $I_2$ and $I_3^-$ from both vapor and aqueous medium (Fig. 1). The strategic embedding of free amine groups pendent discrete chemically stable cationic MOP with a imine functionalized non-interpenetrated 2D COF with 1D open channeled dual-pores, showcased unique structural features with multifunctional properties including, high crystallinity with large surface area, presence of heteroatoms, highly dense N-containing moieties, e.g., imine, amine groups, cationic nature with free $Cl^-$ ions, $Cp_3Zr_3O(OH)_3$ SBU, and low density ultralight aerogel scaffold (Fig. 1). Owing to these multifunctional characteristics, IPcomp-7 displayed efficient sequestration efficiency in terms of high adsorption capacity, fast kinetics, superior selectivity, retention, recovery and recyclability towards $I_2$ and polyiodides under static and dynamic conditions at different temperatures in both vapor and aqueous medium. Moreover, the hybrid composite exhibited rapid and highly selective enrichment of both molecular iodine and polyiodine anions in the presence of a large excess of other interference from different challenging water systems. Consequently, in order to demonstrate the practical utility, as a proof-of-concept device, IPcomp-7 embedded column breakthrough test manifested cyclic efficient capture and recovery of iodine/polyiodides.

## Results

To construct the host-guest-based unique hybrid composite ionic aerogel material, a cationic MOP and a 2D imine COF was selected as suitable guest and host materials, respectively. At first, the individual components, i.e., the nanosized Zr(IV)-based chemically stable amino-functionalized MOP ({[$Cp_3Zr_3O$-$(OH)_3$]$_4$($NH_2$-$BDC$)$_6$}·$Cl_4$)[47] (Fig. 1, Supplementary Fig. 5) and the tetradentate core based 1D channel-type dual-pore containing hierarchical porous COF[48,49] aerogel[33] (Fig. 1, Supplementary Fig. 6) were synthesized and thoroughly characterized (Supplementary Figs. 7–16). The hybrid material (IPcomp-7) was synthesized via stepwise systematic covalent hybridization synthetic strategy. Initially, the amino-group pendent MOP was functionalized with terepthaldehyde and further reacted with the precursors of the COF, tetraamino-tetraphenylethylene along with excess terepthaldehyde to construct the hybrid wet-gel material (Fig. 1). In the subsequent step, after the reactivation and solvent-exchange process this hybrid wet-gel was subjected to conversion into aerogel by applying supercritical $CO_2$ drying procedure to fabricate a covalently threaded nano-snare (Zr(IV)-MOP) decorated cationic crystalline ultralight hybrid composite aerogel material (Zr(IV)-MOP/COF) (IPcomp-7) (Fig. 1)[33]. The following physical and spectroscopic techniques were employed to characterize the structural and morphological features of the synthesized composite material. Powder X-ray diffraction (PXRD) analysis of IPcomp-7 indicated well-resolved strong diffraction peaks that precisely match with the simulated patterns of the reported COF system, suggesting the highly crystalline nature along with dual-pore structure of the composite (Fig. 2a and Supplementary Fig. 11)[48,49]. The Fourier transform infrared spectroscopy (FT-IR) spectra of IPcomp-7 verified both the formation of the imine COF-based aerogel host-matrix (characteristic peaks at ~1622

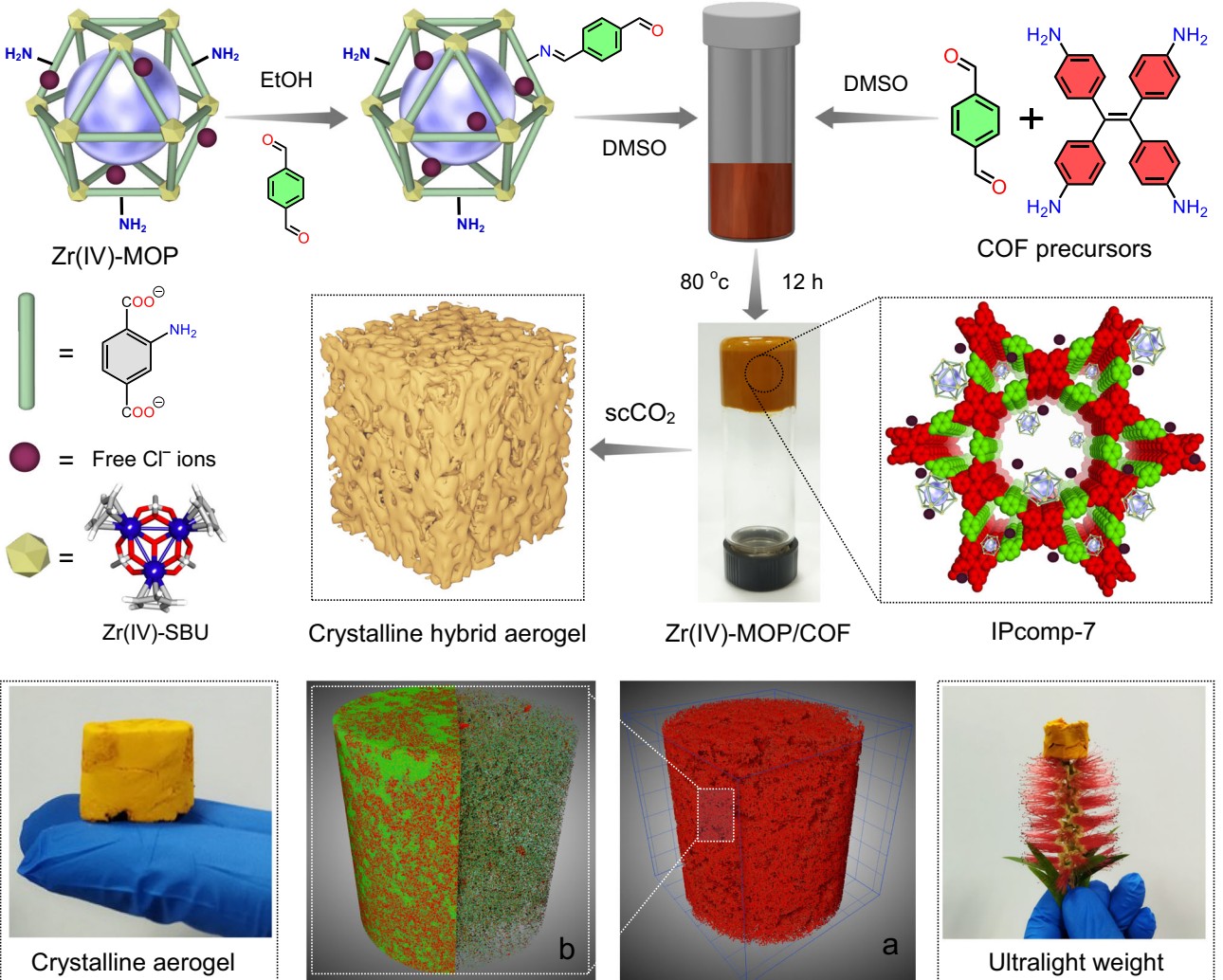

**Fig. 1 | Schematic representation of fabrication procedure and various characteristics of the composite.** Synthesis of NH₂-Zr(IV)-MOP embedded COF hybrid aerogel (IPcomp-7). IPcomp-7 aerogel is crystalline and light weight in nature. The schematic illustration of the MOP exhibiting Zr(IV)-SBU, free Cl⁻ ions and 2-aminoterepthalate ligand. **a, b** 3D X-ray tomographic images of the hybrid aerogel showing presence of hierarchical macropores throughout the structure. (Color code: zirconium: dark blue, oxygen: red, carbon: gray, hydrogen: white).

and ~1693 cm⁻¹, corresponding to the appearance of C=N stretching vibration band, and the attenuation of aldehyde frequencies, respectively)[47–49] and the presence of guest amino-functionalized Zr(IV)-MOP (characteristic peaks at ~1380, ~3515, ~3390 cm⁻¹, corresponding to metal-carboxylate (Zr-O) bonds, symmetric and asymmetric stretching frequencies of free -NH₂ group, respectively[50] in the hybrid composite material (Supplementary Fig. 17). This data evidenced the presence or grafting of NH₂-MOP into the COF aerogel matrix along with exposed Zr(IV)-SBU and excessive free amino functional groups. Furthermore, as compared to pristine COF aerogel, the X-ray photoelectron spectroscopy (XPS) survey spectra of IPcomp-7 showed peaks related to elements Zr and Cl along with C, N, and O, indicating successful implantation of cationic MOPs into the COF aerogel matrix (Fig. 2b). Also, the Zr *3d* XPS spectra of the pristine MOP was found to shift slightly after binding with COF-aerogel, indicating significant interaction between host and guest (Fig. 2c). The elevated thermal behavior of the MOPs in the composite was observed by the thermogravimetric analysis (TGA) of IPcomp-7, which also displayed features of both MOPs and COF aerogel (Supplementary Fig. 18). In addition to this, solid-state ¹³C cross-polarization magic angle spinning (CP MAS) nuclear magnetic resonance (NMR) spectroscopy analysis was performed in order to structurally characterize the types of

molecular level interactions that occur between the host-COF matrix and guest-MOP molecules in IPcomp-7. The presence of distinct peaks in the spectra of IPcomp-7 demonstrates that the hybrid composite contained both MOP and COF functionalities (Supplementary Fig. 19). Additionally, the appearance of a new characteristic signal at ~171 ppm in the spectra of IPcomp-7 indicated the formation of imine bond between the amine group of MOP and aldehyde moiety of COF[51,52].

Following this, NMR, high-resolution mass spectrometry (HRMS), and inductively coupled plasma (ICP) experiments have been carried out to verify the presence, as well as the leaching test of guest MOP molecules from IPcomp-7 (Supplementary Note 3)[47,50]. Afterwards, the surface morphological investigation of IPcomp-7 was performed by field emission scanning electron microscopy (FESEM), transmission electron microscopy (TEM), and confocal laser microscopy. It was discovered from the FESEM images of IPcomp-7, the morphology resembled sponge-like architecture (Fig. 2d, Supplementary Fig. 24)[33]. Moreover, the high-magnification FESEM images of MOP-embedded hybrid COF aerogel revealed a disordered pattern of interconnecting fibers with meso-macro hierarchical large void spaces (~5-50 μm), which suggested the 3D hybrid aerogel was highly porous in nature. Analysis of TEM images revealed the interconnected agglomerated particles with irregular large voids caused by the random stacking of

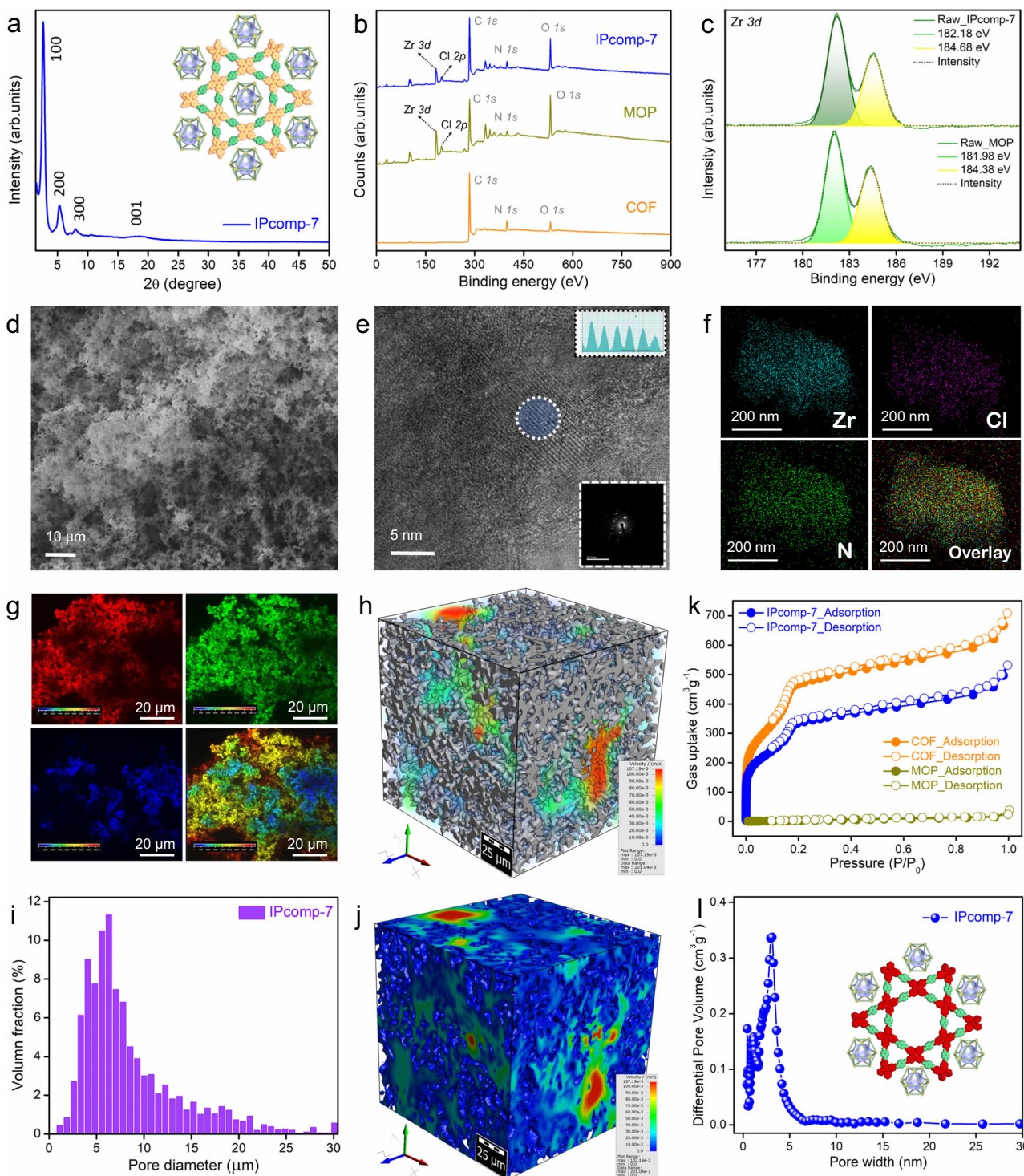

**Fig. 2 | Structural and morphological characterizations of the composite.**
**a** PXRD pattern of IPcomp-7. **b** XPS survey spectra of IPcomp-7 with MOP and COF.
**c** Zr *3d* XPS spectra for MOP and IPcomp-7. **d** FESEM image, **e** HRTEM with
d-spacing and FFT image, **f** HAADF-TEM elemental mapping of the composite.
**g** Color-coded 3D confocal microscopic image of IPcomp-7. **h** Color-scale
visualization of 3D X-ray tomographic image with pore-size distribution of IPcomp-7 (grey color represents solid volume). **j** Void volume of IPcomp-7 with color-scale (blue to red color represents void volume). **k** N$_2$ gas sorption data. **i, l** Micron size and NLDFT pore distribution histogram of IPcomp-7, respectively.

the fiber-like nanolayers (Supplementary Fig. 25). Further, to gain insight into the nanostructure of the hybrid aerogel we closely analyzed the high-resolution transmission electron microscopy (HRTEM) images of IPcomp-7. The HRTEM images displayed a characteristic interplanar spacing of 0.23 nm, which is in good agreement with the d-spacing of the COF structure (Fig. 2e)[48,49]. This observation

established the highly ordered/crystalline nature of the MOP loaded hybrid COF aerogel system (fast Fourier transformation, (FFT) data, inset of Fig. 2e)[29,53]. A detailed examination of the FESEM and TEM energy-dispersive X-ray (EDX) data of IPcomp-7 revealed the element Zr/Cl ratio ~3.0, which is in agreement with the crystallographic composition of the Zr(IV)-MOP[47], supported the stable existence of the

cationic MOPs in the hybrid COF aerogel (Supplementary Tables 1–6). Moreover, high-angle annular dark-field (HAADF) TEM and FESEM elemental mapping demonstrated the consistent homogenous distribution of all pertinent elements of the cationic MOPs throughout the structure of the composite (Fig. 2f, Supplementary Figs. 26–29). Furthermore, the confocal fluorescent microscopic 3D imaging of IPcomp-7 indicated plenty of extrinsic micron-sized pores (macropores) with weakly interconnected nanoparticles surface morphology (Fig. 2g, Supplementary Fig. 30).

Additionally, microscale X-ray computed tomography (CT) experiments were carried out to investigate the 3D macroscopic hierarchical porous structure of the hybrid nanocomposite. The 2D cross-sectional CT images of the hybrid aerogel material was discovered to include a heterogeneous distribution of large micron-sized pores (~1–20 μm) throughout the structure (Supplementary Fig. 31). The segmented 3D models of IPcomp-7 were created using reconstruction of 2D CT images for improved comprehension. A disordered pattern of interconnecting macropores was visible across the volume of IPcomp-7 in the segmented 3D CT image (Fig. 2h, Supplementary Fig. 32). Also, the corresponding color-coded 3D CT images made it easy to recognize the large pore size distribution (PSD) (high volume fraction within ~1–20 μm with maximum size of 5–7 μm) and 80.2 ± 2% void space inside the entire volume of IPcomp-7 (Fig. 2i, j, Supplementary Fig. 33). These findings demonstrated the presence of extrinsic macropores throughout the nanocomposite together with 3D morphological characteristics, all of which are advantageous for rapid mass transfer within porous materials along with easy and complete accessibility of functional sites, and shortened diffusion paths, beneficial towards effective iodine adsorption[54].

Furthermore, the micro- and mesoporosity of IPcomp-7 was examined using low-temperature (77 K) $N_2$ gas sorption studies. A sharp uptake at low pressure (below $P/P_0 = 0.01$) (type I) with a step at $P/P_0 = 0.15–0.20$ bar pressure (type IV) were identified from the sorption data of IPcomp-7, demonstrating the presence of both microporous and mesoporous features, respectively (Fig. 2k)[38,55,56]. A similar observation was noted in the case of pristine COF-aerogel (Fig. 2k). However, it was found that compared to the bare COF-aerogel, IPcomp-7 exhibited significantly lower gas uptake or less porosity. This phenomenon can be explained due to the accommodation of nonporous guest-MOP molecules into the hierarchical porous COF-aerogel matrix. The Brunauer−Emmett−Teller (BET) surface area of IPcomp-7 was calculated to be 1463 $m^2g^{-1}$. The pore size distribution was calculated using nonlocal density functional theory (NLDFT) methods revealed the presence of micropores (<2 nm) and mesopores (>2 nm) within the hybrid material (Fig. 2l). All these experiments collectively confirmed that IPcomp-7 exhibited hierarchical porosity with micropores (<2 nm), mesopores (>2 nm) and macropores (~5 μm). Other optical characterizations also indicated the successful formation of the MOP/COF hybrid composite (Supplementary Note 4)[51,55]. To our delight, in spite of their low density and high porosity, IPcomp-7 demonstrated excellent mechanical flexibility, making them sturdy enough to be used in sequestration applications[33].

## Iodine sequestration study

Initially, we examined the optimal amount MOPs utilized in the hybrid composite synthesis that enabled the highest efficacy toward iodine vapor capture under optimized conditions. It was found that the composite synthesized by taking ~15 mg of MOP molecules gave the best result (Supplementary Fig. 38). We postulated that using excess MOPs lead to the inevitable aggregation through the host matrix's interconnected hierarchical framework, which reduces porosity and renders a larger number of active sites of the overall nanocomposite inaccessible. This ultimately resulted in lowering of the $I_2$ sequestration efficiencies. The static iodine capture performance of the nano-adsorbent (IPcomp-7) along with the pristine

materials was investigated following a typical procedure. Interestingly, the color of the samples became dark-black upon exposure of iodine indicating excess occlusion of iodine into the porous structure of the hybrid composite (Supplementary Fig. 39). From the kinetics experiment, it was found the maximum or equilibrium adsorption capacity of IPcomp-7 was 9.98 $g.g^{-1}$ within 24 h of operation, with a sequestrate step of 7.87 and 9.18 $g.g^{-1}$ in 6 and 12 h, respectively (Fig. 3a). The maximum capacity of IPcomp-7 was found to be much higher than the pristine MOP (2.31 $g.g^{-1}$) and COF-aerogel (6.11 $g.g^{-1}$) materials, which clearly indicated the advanced role of the multifunctional optimum hybrid composite over individual materials towards efficient sequestration of gaseous iodine (Fig. 3a). It should be mentioned that the static iodine capture capacity of IPcomp-7 at 75 °C stand as one of the highest among most of other reported iodine adsorption materials (Supplementary Table 7). The iodine adsorption kinetic of IPcomp-7 was found to follow pseudo-second-order model, which explored the kinetic rate constant as high as 0.0567 $g.g^{-1}h^{-1}$ (Supplementary Fig. 40). This value suggested the rapid iodine sequestration rate of IPcomp-7, owing to its macro-micro hierarchical interconnected low-density 3D porous structure. Furthermore, considering the industrially relevant conditions, the iodine capture experiment has been performed at different temperatures, such as 25 and 150 °C in both dry and humid condition. The result indicated 4.27 and 2.89 $g.g^{-1}$ iodine capture capacities at 25 and 150 °C temperature, respectively, at dry condition (Fig. 3b). In addition, under humid condition, only a small reduction in the adsorption capacities of IPcomp-7 were recorded (Fig. 3b). These results demonstrates that humidity had a negligible impact on the ability to efficiently extract iodine from vapor by IPcomp-7. That being said, reusability is an important parameter for an ideal adsorbent towards effective iodine sequestration. Therefore, recyclability of IPcomp-7 was evaluated by performing the desorption test of iodine laden composite. The desorption test resulted ~97% iodine release efficiency in the first cycle. IPcomp-7 demonstrated high sorption capacity (>7.61 $g.g^{-1}$) even after five consecutive capture-release cycles indicating superior reusability of the material (Supplementary Fig. 41). The release of captured iodine was also performed by treating the $I_2$@IPcomp-7 with hexane solution. Based on the UV-vis data, the concentration of iodine in n-hexane increased as the treatment progressed, which resulted >96% release of iodine within 2 h of contact time (Supplementary Figs. 42, 43). In this way, IPcomp-7 can be reused for iodine vapor capture, which demonstrated improved recyclability up to five cycles maintaining high efficiency and capacity. Apart from recyclability, another important parameter for an ideal iodine sorbent is retention efficiency. IPcomp-7 exhibited very high adsorption stability (>94% up to 7 days) with minimal liberation of iodine, indicating strong binding affinity of iodine with the nano-adsorbent (Fig. 3c). However, in comparison to IPcomp-7, the retention efficiency of the pristine COF-aerogel and MOP was found to be significantly lower.

Pertaining to the practical implementation of the material towards efficient iodine adsorption application, dynamic vapor phase iodine sorption experiments were carried out following the standard procedure as described in the supporting information file (Supplementary Figs. 44, 45). The dynamic iodine adsorption test at 75 °C of IPcomp-7 resulted 3.76 $g.g^{-1}$ capture capacity (Fig. 3d). Also, the kinetic constant of IPcomp-7 for dynamic iodine capture was calculated to be high 0.0627 $g.g^{-1}h^{-1}$ indicating fast diffusion of iodine within hierarchical pores of hybrid aerogel (Supplementary Fig. 46). However, when compared to IPcomp-7, the pristine COF-aerogel and MOP exhibited significantly lower dynamic iodine adsorption capacity. Moreover, the recyclability test of IPcomp-7 under dynamic condition was elucidated. As was observed, after five capture and release cycles, the iodine capacity of IPcomp-7 was found to be more than 2.48 $g.g^{-1}$ (Supplementary Fig. 47). After being exposed to different radiation

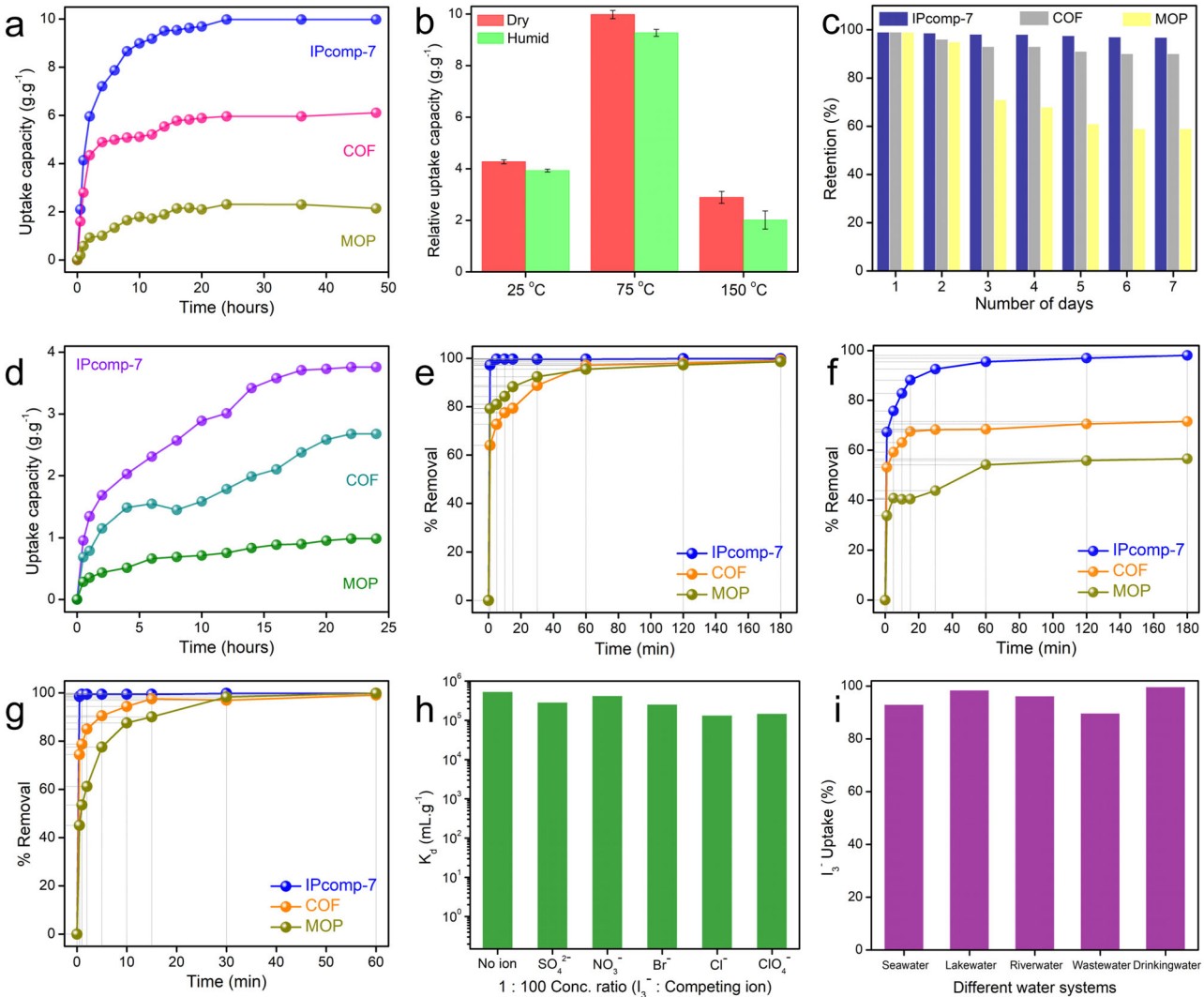

**Fig. 3 | Result of vapor and aqueous phase iodine/polyiodide sequestration studies. a** Gaseous iodine uptake capacities of IPcomp-7, COF and MOP in static system at 75°C. **b** Relative static iodine uptake capacities of IPcomp-7 at different temperatures (The error bars are the standard deviations from three parallel measurements). **c** Iodine retention efficiency of IPcomp-7, COF and MOP. **d** Iodine uptake capacities of IPcomp-7, COF and MOP in dynamic condition at 75°C. Iodine removal efficiency of IPcomp-7, COF and MOP in **e** water as $I_2$, **f** n-hexane, **g** water as $I_2/KI$. **h** Distribution coefficient ($K_d$) profile of IPcomp-7 for $I_3^-$ uptake. **i** Relative $I_3^-$ uptake efficiency by IPcomp-7 in different water systems.

doses, the composite also demonstrated comparable high iodine absorption capabilities in the static vapor phase condition, suggesting radiation resistance ability (Supplementary Fig. 48).

Notably, the majority of the described iodine uptake experiments only demonstrated vapor-phase iodine extraction, and primarily examined the exchange capacity. Yet, despite being crucial for real-time water treatment applications, selective and speedy extraction of iodine from aqueous-phase low concentrations in the presence of a significant excess of other interfering anions has received scant attention. Therefore, the ability of iodine uptake from its aqueous solution was evaluated with IPcomp-7 along with the pristine MOP and COF-aerogel materials. The UV-vis studies indicated the almost instantaneous diminishing of iodine concentration from the saturated aqueous solution upon treating with IPcomp-7 (Supplementary Fig. 49). However, the decrement in the intensity/concentration of iodine by MOP and COF-aerogel was relatively less compared to IPcomp-7 (Supplementary Figs. 50–53). This result demonstrated the rapid uptake kinetics (~99.3% removal within 1 min) of molecular iodine from water by IPcomp-7 (Fig. 3e). Conversely, the iodine uptake efficiency of pristine MOP and COF-aerogel was found to be inferior to

that of IPcomp-7 (Fig. 3e). The aqueous phase iodine removal kinetics of IPcomp-7 was found to follow pseudo-second–order kinetics model (Supplementary Figs. 54, 55). Moreover, the iodine uptake capacity of IPcomp-7 from water was calculated to be 4.74 g.g⁻¹. In addition, IPcomp-7 was further applied for sequestrate iodine from n-hexane medium, which demonstrated highly efficient capture as ~97.9% removal of iodine was observed from the stock solution within 60 min of contact time (Fig. 3f, Supplementary Fig. 56). Whereas, relatively lower efficiency in iodine uptake was found in case of pristine MOP and COF-aerogel (Fig. 3f, Supplementary Figs. 57–60). The rate constant and Langmuir model-based maximum iodine uptake capacity of IPcomp-7 in n-hexane was 0.00885 g.mg⁻¹.min⁻¹ and 4.23 g.g⁻¹, respectively (Supplementary Figs. 61, 62). Furthermore, considering the cationic nature of IPcomp-7 with free exchangeable chloride anions, sequestration of iodine in the form of polyiodide species (such as $I_3^-$) from water was performed by treating IPcomp-7 with a concentrated aqueous solution of KI and $I_2$. From the time-dependent UV-vis studies, it was found that IPcomp-7 exhibited ultrafast iodide ($I_3^-$) removal kinetics (>99% removal within 1 min) from KI/$I_2$ solution, whereas, pristine materials showed a relatively sluggish uptake rate

(Fig. 3g, Supplementary Figs. 63–67). The $I_3^-$ uptake process by IPcomp-7 found to follow pseudo-second-order kinetics model with rate constant 1.9437 g.mg$^{-1}$.min$^{-1}$, which suggested the ultrafast capture efficiency (Supplementary Fig. 68). Moreover, the concentration-dependent equilibrium uptake capacities of KI/$I_2$ were fitted well with the Langmuir isotherm model, which determined the maximum iodine sorption capacity of IPcomp-7, was 5.16 g.g$^{-1}$, which is one of the highest value to that of other reported materials (Supplementary Fig. 69 and Supplementary Table 8). Furthermore, the distribution coefficient ($K_d$) value is an important parameter, which suggests the affinity of a sorbent material toward the sorbate, and $K_d > 10^4$ mL.g$^{-1}$ is considered as exceptional[12]. Hence, the $K_d$ value for $I_3^-$ capture by IPcomp-7 was also investigated and calculated to be in the range of ~$10^6$ mL.g$^{-1}$, indicating a significant affinity for $I_3^-$ in water (Fig. 3h).

Henceforth, considering the real-world applications, such as detoxification of groundwater, an ideal adsorbent should selectively remove targeted contaminants in the presence of a large excess of other interferences in water. Typically, anions like $NO_3^-$, $Cl^-$, $SO_4^{2-}$, $Br^-$ etc. are present in significant excess amounts with $I_3^-$ contaminated water, which negatively impacts the adsorbent's overall sorption effectiveness[56]. Therefore, to test the binary and mixture selectivity, $I_3^-$ capture study has been performed in the presence of equimolar as well as ~100-fold excess other competing anions. IPcomp-7 demonstrated highly efficient extraction of $I_3^-$ with removal efficiency of >97.9% in case of both equimolar and ~100-fold excess anions (Supplementary Figs. 70–72). Moreover, the corresponding $K_d$ values for all the binary/mix-competing mixture were evaluated and were found in the order of ~$10^5/10^6$ mL.g$^{-1}$, advocating exceptional binding affinity of IPcomp-7 towards $I_3^-$ (Fig. 3h). Motivated by such highly efficient and selective entrapment of $I_3^-$ anions by IPcomp-7 in water, further sequestration of $I_3^-$ from different complex water matrixes, including seawater, lake-water, river-water, etc. have been performed. Interestingly, in all the water systems, the hybrid composite exhibited almost >85% removal efficiency of $I_3^-$ anions within 1 min (Fig. 3i). Additionally, the maximum

capture capacity and $K_d$ value of IPcomp-7 for $I_3^-$ in seawater was calculated as 5.09 g.g$^{-1}$ and ~$10^5$ mL.g$^{-1}$, respectively. Given the seawater contain large excess of competing ions and fouling agents, these values are extraordinarily high.

Inspired by the ultrafast and efficient sequestration of molecular iodine and triiodides from water, we sought to explore further the potential of the hybrid composite toward dynamic column-based aqueous phase iodine capture experiments. Initially, to test the flow-through-based iodine capture from water, a column was packed with IPcomp-7 and aqueous iodine solution having range of different concentration was passed through it (Fig. 4a, Supplementary Fig. 73). The initial dark/light-yellow iodine solution eventually turned colorless after passing through the column. Additionally, the UV-vis analyses indicated that IPcomp-7 was able to capture more than ~96% of iodine from water across all the concentrations (Fig. 4b, Supplementary Fig. 74). Furthermore, we were encouraged to perform the aqueous phase $I_3^-$ breakthrough adsorption experiment with IPcomp-7 embedded column as schematically represented in Fig. 4c, Supplementary Fig. 75. In this typical experiment, the column was charged with aqueous solution of $I_2$ + KI along with other competing anions with a flow-rate of 0.5 mL.min$^{-1}$, and the concentration of the column-eluted solution was determined by UV-vis analyses.

Noteworthy, almost >99% elimination efficiency of triiodide was observed passing >200 mL of solution in the first cycle of the break-through test (Fig. 4d). Additionally, the column's capability toward regeneration was examined using aqueous NaCl and hexane solution, which revealed notable removal efficiency with admirable uptake capacity over the course of three cycles (Fig. 4d, Supplementary Fig. 76). In view of practical implication, these results demonstrated impressive segregation effectiveness of IPcomp-7 packed column towards iodine in both environmental and industrially contaminated water. Such selective, regenarable and large iodine uptake capacities of the overall nanocomposite is believed to an outcome of the cooperative effect of Zr(IV)-SBU, free -$NH_2$ groups, cationic nature with

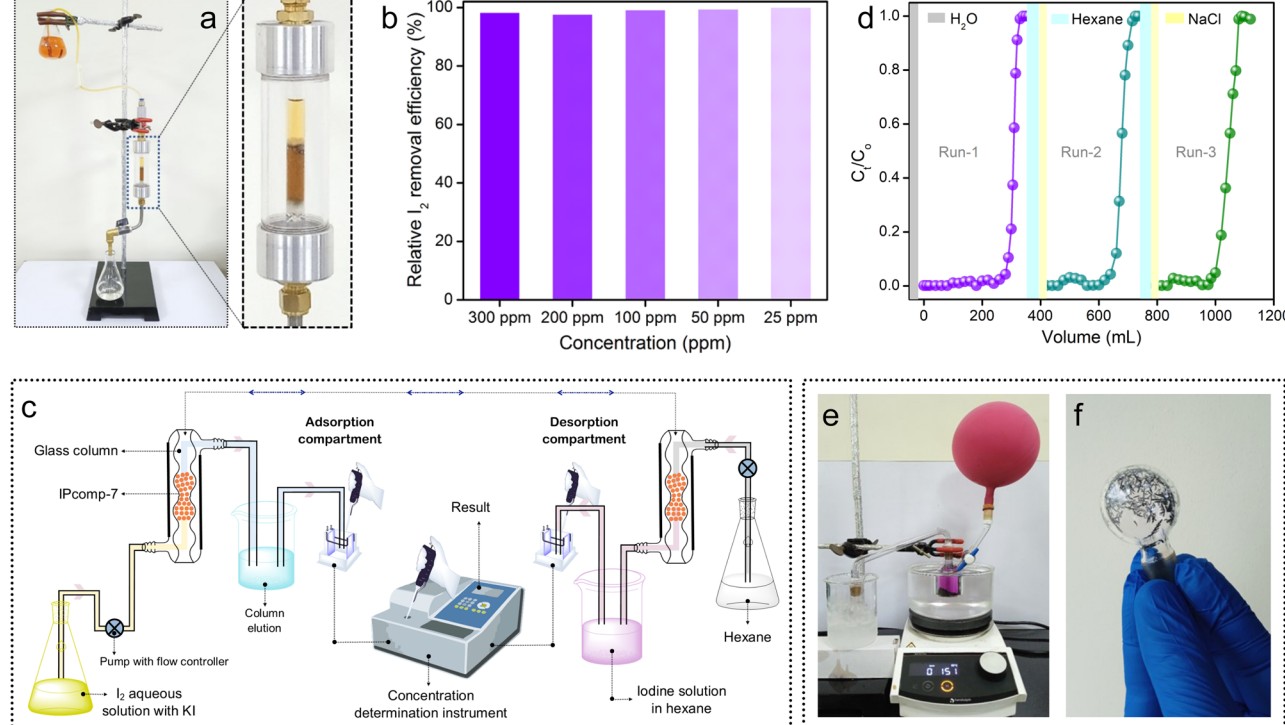

**Fig. 4 | Result of aqueous phase iodine/polyiodide dynamic sorption and recovery studies. a** Digital image of dynamic column-based aqueous phase molecular iodine capture test. **b** Result of dynamic column-based $I_2$ capture experiment. **c** Schematic diagram of IPcomp-7 embedded column-based flow-through $I_2$/KI capture study. **d** Breakthrough profile of column-based $I_2$/KI capture test. **e, f** Images of solid iodine recovery test set-up and recovered iodine crystals.

exchangeable Cl⁻ ions of the guest-MOP molecules combined with heteroatom functionality, hierarchical macro-micro porosity with a large surface area of low-density COF-aerogel matrix. Moving ahead, we performed the recovery test of solid iodine both from vapor and organic solvent medium as described in the supplementary file. In a typical experimental setup, vapor-phase iodine loaded compound was heated at a specific temperature to produce substantial amounts of solid molecular iodine. (Fig. 4e, f, Supplementary Fig. 77). Moreover, the compound-embedded iodine was also recovered by treating the I$_2$@IPcomp-7 with pentane solvent (Supplementary Fig. 78). The rapid evaporation of pentane at room temperature afforded a large quantity of iodine crystals (Supplementary Fig. 79). These experiments demonstrated the facile and potential recovery of iodine form vapor phase by IPcomp-7.

### Investigation of the iodine adsorption mechanism

We were inspired to investigate the adsorption mechanism of IPcomp-7 because of its exceptional iodine capture performance. To understand the iodine sorption mechanism, a series of experiments were implemented on I$_2$@IPcomp-7. FESEM images of IPcomp-7 taken before and after I$_2$ absorption show nearly unaltered morphologies, demonstrating high stability of the material (Supplementary Fig. 80).

The homogenous iodine distribution over IPcomp-7 was revealed by FESEM-EDX data and elemental mapping images, which implies that excessive I$_2$ was absorbed by the hybrid composite (Supplementary Figs. 81, 82). The X-ray 3D CT images of I$_2$@IPcomp-7 indicated the retention of micron-sized macroscopic hierarchical porous structure (Supplementary Fig. 83).

Moreover, the pore size distribution with high volume percentage within ~1 to ~20 μm of IPcomp-7 after the iodine adsorption were clearly visible on the accompanying color-coded 3D CT images (Fig. 5a, Supplementary Fig. 84). The fluorescence microscopic imaging of I$_2$@IPcomp-7 displayed significant quenching of intensity after the iodine adsorption (Supplementary Fig. 85). Next, the time-resolved FT-IR spectra of iodine treated compound indicated obvious shift or reduced intensities of C = N bond stretching vibration from ~1622 cm⁻¹ for IPcomp-7 to ~1631 cm⁻¹ for I$_2$@IPcomp-7 (Fig. 5b)[24]. Similar diminished intensities in the vibration bands of C = C, C-H, and C-N bonds of phenyl rings at ~1556, ~1410 and ~1196 cm⁻¹ were observed. In addition, the bands associated with the Zr-metal-carboxylate (Zr-O) bonds at ~1380 cm⁻¹ of the MOP underwent a gradual shift after adsorbing iodine. All these spectral changes suggested the occurrence of strong charge-transfer-induced interaction between the absorbed iodine molecules and the multiple functionalities, including imine, amine,

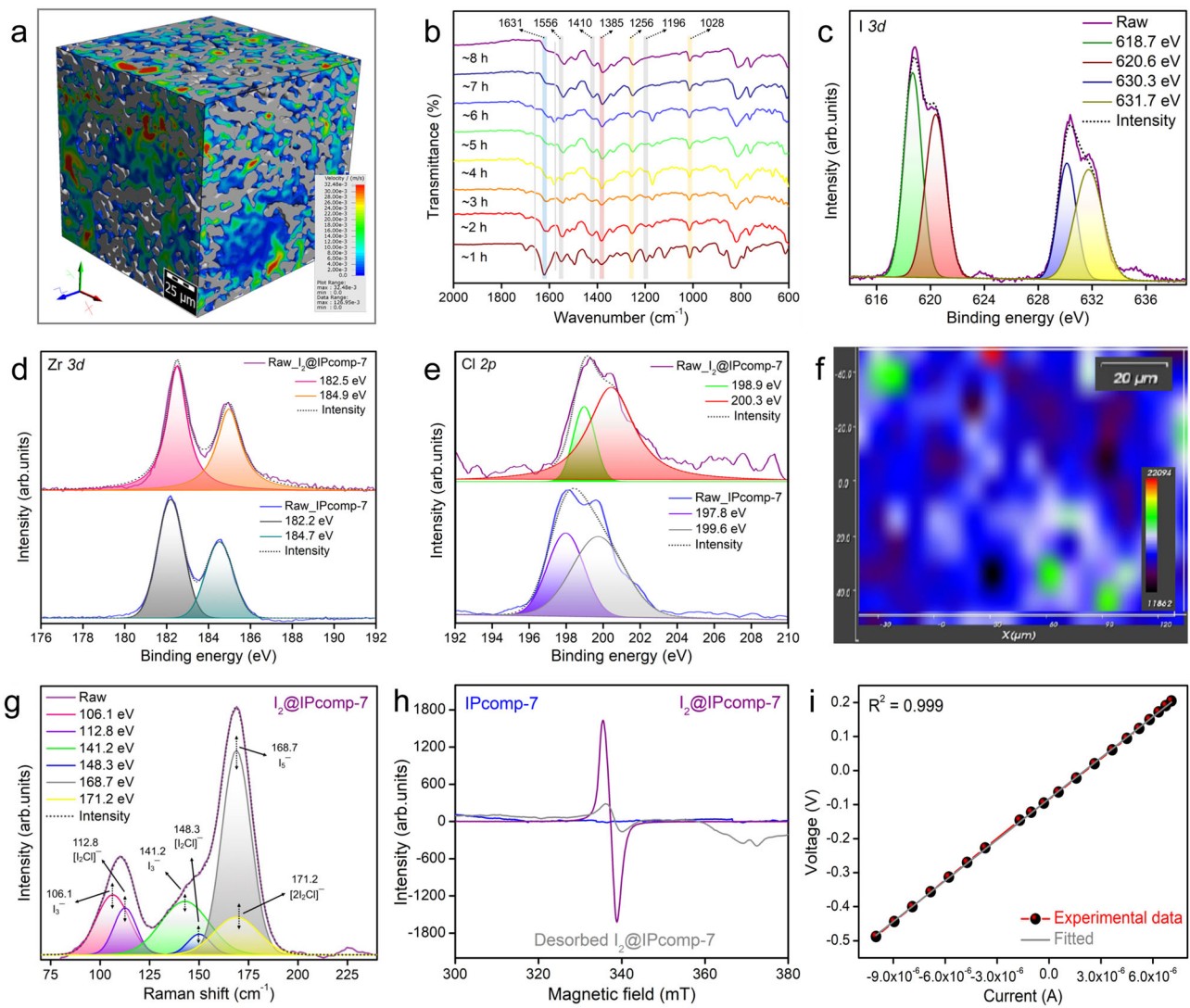

**Fig. 5 | Post iodine adsorption characterizations. a** Color-scale 3D CT image of I$_2$@IPcomp-7. **b** Time-dependent FT-IR spectra of I$_2$@IPcomp-7. **c** I *3d*, **d** Zr *3d*, **e** Cl *2p* XPS spectra of IPcomp-7 before and after iodine adsorption. **f** Raman mapping, **g** Raman spectra of I$_2$@IPcomp-7. **h** EPR spectra of IPcomp-7, I$_2$@IPcomp-7 and desorbed I$_2$@IPcomp-7. **i** Current-voltage (I-V) plots of I$_2$@IPcomp-7.

phenyl and other heteroatoms of the composite[25,29]. Thereafter, XPS was employed to investigate in detail the existing state of trapped iodine as well as the binding nature of iodine/polyiodide anions with the hybrid composite. As compared to IPcomp-7, the XPS survey spectra of I₂@IPcomp-7 clearly indicated the appearance of characteristic peaks for iodine (Supplementary Fig. 86). The I $3d$ XPS spectra of I₂@IPcomp-7 displayed two noticeable peaks located at 618.7 and 630.3 eV associated to I $3d_{5/2}$ and I $3d_{3/2}$ orbitals of iodine molecules respectively, which further indicated that the adsorbed iodine partially exists as molecule (Fig. 5c)[25,57]. There were also two additional peaks, with energies of 620.6 and 631.7 eV, which are ascribed to the generation of polyiodide anions like $I_3^-$ and $I_5^-$[14,57,58]. Also, after iodine adsorption, the N $1s$ spectra of IPcomp-7 was found to shift from 398.7 eV and 399.5 eV to 399.1 eV and 399.9 eV, respectively, indicating interaction of iodine with heteroatom-N (from imine and amine moieties) of the composite (Supplementary Fig. 87). In addition, the emergence of a new peak fraction at 401.2 eV related to the N-I bond revealed the formation of charge-transfer complexes between I₂ and N-center of IPcomp-7[26,29]. Importantly, the Zr $3d$ XPS peaks of I₂@IPcomp-7 were observed to shift towards relatively higher binding energy, suggesting the potential interaction of iodine with Zr-SBU of the MOP of hybrid composite (Fig. 5d)[19]. Moreover, after I₂ adsorption on IPcomp-7, the peak of the Cl $2p$ core energy level shifted from 197.8 and 199.6 eV to 198.9 and 200.3 eV, respectively, suggesting that Cl⁻ was also engaged in the I₂ uptake. (Fig. 5e)[5,9]. All these results demonstrated that along with the COF matrix, the amino-functionalized cationic Zr(IV)-MOP actively participates in the iodine sorption process. Further, the Raman spectra of I₂@IPcomp-7 showed distinct bands at ~106.1, ~141.2, and ~168.7 cm⁻¹ following iodine adsorption, indicating strong charge-transfer interaction of iodine with IPcomp-7 and generation of a multitude of anionic polynuclear iodide species (Fig. 5g, Supplementary Fig. 88)[14]. Among them, band at ~106.1 and ~141.2 cm⁻¹ can be assigned to the symmetric and asymmetric stretching vibration of $I_3^-$, respectively, whereas, the band at 168.7 cm⁻¹ is attributed to the stretching vibration of $I_5^-$ polyiodide species[14,25,26,29,57,58]. In addition to this, peaks associated with symmetric and asymmetric stretching vibration bands of $[I_2Cl]^-$ at 112.8 and 148.3 cm⁻¹, respectively, and stretching vibration bands of $[2I_2Cl]^-$ at 171.2 cm⁻¹ were observed, which indicated potential interaction of iodine with free Cl⁻ anions of the composite[5,25].

Moreover, the spatial distribution of iodine species throughout the structure of IPcomp-7 was investigated by Raman spectroscopy mapping analysis. The color-coded Raman mapping of I₂@IPcomp-7 exhibited a distribution of absorbed iodine species with significant intensities of respective bands (Fig. 5f, Supplementary Fig. 89). The solid-state UV-vis spectra of I₂@IPcomp-7 displayed characteristic broad absorption peak, which indicated the formation of charge-transfer complexes between polyiodide anions and IPcomp-7 (Supplementary Fig. 90)[14,59]. Further, to confirm the existence of such charge transfer interactions, electron paramagnetic resonance (EPR) studies was conducted. The iodine-loaded composite revealed sharp EPR signals, however, almost no paramagnetic signals were observed in case of pristine IPcomp-7 (Fig. 5h). Importantly, the EPR signal was noticeably less after desorption of iodine from IPcomp-7. These findings compel us to hypothesize that I₂@IPcomp-7 contains radicals, which most likely originate from an electron transfer between the electron-rich moieties of IPcomp-7 and entrapped iodine species[14]. Also, notably following the capture of I₂, the signal in the solid-state ¹³C CP-MAS NMR spectra of I₂@IPcomp-7 clearly broadens (Supplementary Fig. 91). This may result from a strong affinity between the π-electron-rich moieties of IPcomp-7 and the electron-deficient iodine molecules[26,29]. The PXRD patterns of I₂@IPcomp-7 exhibited no diffraction peaks, indicated loading of amorphous iodine in the pores of the composite as well as eliminating the possibility of deposition or recrystallization of I₂ on the surface of IPcomp-7 (Supplementary Fig. 92)[26,29]. The TGA profile of I₂@IPcomp-7

indicated high weight loss, demonstrating large quantity of iodine occupied into the pores of IPcomp-7 (Supplementary Fig. 93). Furthermore, the electrical conductivity (σ) of the iodine-loaded composite was measured, which yielded bulk σ value of ~7.83 × 10⁻⁵ S.cm⁻¹ (Fig. 5i). This value suggested the successful occupation of excess iodine or polyiodide anions into the pores of IPcomp-7[12,20]. Moreover, in order to investigate the role of NH₂-MOP towards iodine adsorption, ¹H NMR time-dependent titration experiments of I₂@NH₂-MOP were performed, which disclosed the significant degree of shift in the signals for the protons of the respective moieties (Supplementary Fig. 94). This shift in the signals demonstrated that these moieties had a preferential location for interacting with iodine species. All of these findings demonstrated that the contribution of both physisorption and chemisorption were at play in tandem to selectively adsorb the iodine species by the different multiple functionalities, including heteroatomic sites, amine, imine, phenyl moieties, Zr(IV)-SBU, free Cl⁻ counter-anions, etc. of IPcomp-7.

We were also encouraged to disseminate the advancement of low-density hybrid aerogel with hierarchical micron-sized interconnected macropores towards the ultrafast iodine capture kinetics of IPcomp-7 in both vapor and aqueous phase. For this, confocal microscopic experiment and flow velocity analysis of iodine-contaminated vapor and water was estimated by numerical simulations studies (Supplementary Note 5). The 3D confocal images of IPcomp-7 showed the existence of large number of open voids (macropores) throughout the surface (Supplementary Fig. 95). These large open pores enable easy encapsulation of iodine molecules or polyiodide anions through the interconnected pore channels, which allow continuous rapid mass flow. The 2.5D confocal images shown were acquired from several layers of the composite. The modified 2.5D images of different layers (5th, 25th, 45th, and 65th) displayed dispersive adsorption of analyte within the different 2D layers of the whole matrix of IPcomp-7, as opposed to only the surface of the composite. This supports the easy and fast penetration of water or vapor through the open pores of IPcomp-7 further promotes the excessive adsorption of iodine and polyiodide anions (Fig. 6a–d, Supplementary Fig. 96).

Moreover, in order to make an evidence-based forecast of the rapid and dynamic events occurring throughout the porous structure towards I₂/I₃⁻ entrapment process in vapor and water, an *in-silico* mass transport simulation was carried out in the segmented 3D tomographic model of IPcomp-7 (Supplementary Note 5). Using an appropriate numerical solver at 25 °C, 20 Pa pressure-drop, etc., a simulation study was performed to visualize flow in the vertical axis (Y axis), and the outcome visualizes the average flow velocity profiles. The findings of the numerical simulation method forecast the practicable average flow profiles through the inter-connected void-space of IPcomp-7 (Fig. 6e). The simulation results indicated entrapment of vapor through the porous structure of the hybrid aerogel. The average flow velocity in Y-direction was calculated as ~4.2 × 10⁻³ ms⁻¹, with flow resistivity of 3.4 × 10⁷ kg/(m³s) and permeability of 5.3 × 10⁻¹³ m², respectively (Fig. 6f–i, Supplementary Figs. 97, 98). The result of water flow-velocity range in the hybrid aerogel demonstrated interconnected hierarchical pores of IPcomp-7 greatly aids in the massive mass-transport for rapid and high iodine uptake efficiency. This study further suggests the significance of low-density aerogel materials for effective iodine sequestration.

The potential of Zr(IV)-SBU and amino functional groups to selectively interact with iodine/polyiodides is well established in the literature[11,19,59,60]. Therefore, the cationic nano-snare (NH₂-Zr(IV)-MOPs) in the hybrid composite effectively contributes towards the selective iodine capture owing to its multifunctional properties. However, the functional groups of the pristine MOPs become inaccessible to interact with I₂/I₃⁻ as a result of inevitable random self-aggregation, which significantly lowers the capture efficiency. Howbeit, in regard to the hybrid composite, the host COF-matrix wired up the MOPs separately, keeping the majority of the active sites well

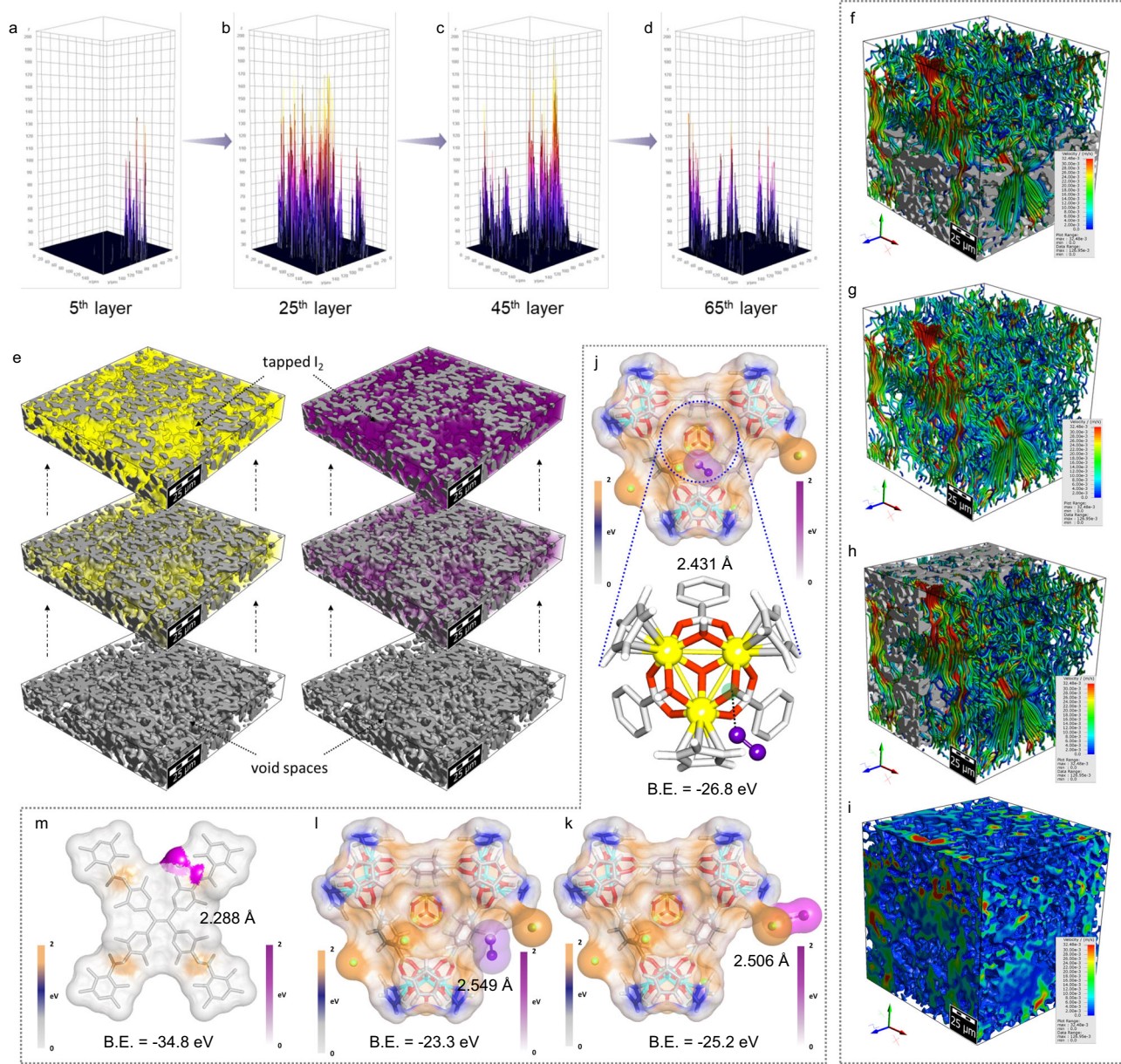

**Fig. 6 | Adsorption mechanism and DFT calculations. a–d** The 2.5D confocal fluorescence microscopic images of IPcomp-7 with different 2D layers. **e** 3D images of IPcomp-7, generated by the numerical solver, depicting the entrapment process of iodine vapors, at three different stages, in the voids of the hybrid aerogel. **f–i** The water flow-velocity profile inside IPcomp-7, estimated by the numerical solver, visualizing the flow paths. **j–m** Energy-optimized structure of various interaction between iodine molecule and **j** Zr-SBU, **k** $Cl^-$ ions, **l** free $-NH_2$ group of the MOP, and **m** imine functional group of the COF shows the possible binding sites. (Color code: zirconium: yellow, oxygen: red, carbon: gray, hydrogen: white, iodine: violet) (B.E.: binding energy).

exposed, facilitating the diffusion of $I_2/I_3^-$ species toward the active nano-snare through the macroscopic hierarchical porosity. All these discussion established the iodine or polyiodide uptake mechanism of IPcomp-7 by the cooperative interactions of multiple functionalities, such as strong interaction between heteroatom containing electron rich moieties of the composite with iodine, H-bonding with free $-NH_2$ group of MOP, energetically favorable binding as well as exchange with free $Cl^-$ ions in vapor and aqueous phase, respectively, and the selective preferential affinity towards $I_2$ with deliberately grafted unique Zr(IV)-SBU of the MOP.

Additionally, the theoretical calculation utilizing the density functional theory (DFT) simulation study was conducted, which corroborated with the experimental findings, further verifying the strong interaction between iodine/polyiodide anions and host-guest of the hybrid composite (Supplementary Note 6). We used TPE backbone of

the COF and crystal structure of the $NH_2$-MOP as model molecules for calculating the binding energies with $I_2/I_3^-$ species. The calculations showed that along with others the imine functionality of COF, binding with hydroxy group (·OH) of Zr(IV)-SBU, free $NH_2$ group and $Cl^-$ ions of the MOPs selectively interacts with iodine with binding energies of −34.8, −26.8, −23.3 and −25.2 eV, respectively (Figs. 6j–m, Supplementary Figs. 99–105). Particularly, the optimized geometries disclosed the selective strong interaction between the terminal −OH group of Zr-cluster and iodine/polyiodide anions, which further donates the maximum negative charges to the adsorbed iodine, indicating the maximum charge separation between the iodine atoms (Fig. 6j, Supplementary Figs. 99-107)[19]. Moreover, the DFT calculation displayed that the adsorption configuration of the $I_2$ molecule interacting directly with $Cl^-$ was found energetically favorable (Fig. 6k)[5]. According to these studies, installation of ionic functionality with free

$Cl^-$ ions, Zr-metal cluster, amino and other electron-rich moieties by covalent grafting of the cationic MOP into the imine-COF-aerogel matrix increases the number and strength of selective binding sites for $I_2/I_3^-$ adsorption by virtue of Coulomb interactions to generate thermodynamically more stable complexes[26].

## Discussion

In summary, remarkable sequestration efficacy towards iodine have been demonstrated by successfully synthesizing a unique crystalline hybrid aerogel material by covalent-linking of nanosized cationic MOPs with a hierarchical porous COF-aerogel matrix, utilizing systematic hybridization synthetic strategy. We intensively investigated the structural, morphological and optical properties of such hybrid aerogel to elucidate detailed insights of the multifunctional composite material. Effective entrapment of iodine being the key targeted criterion, the combined interconnected hierarchical macro-microporosity, low-density ultralight weight, high mechanical stiffness along with deliberately grafted Zr(IV)-SBU and free amino group-based discrete cationic nano-snare (MOP) into a imine-COF aerogel matrix exhibited ultrafast, selective and high enrichment towards iodine uptake. Importantly, the hybrid aerogel illustrated a striking performance towards rapid and selective uptake with high capacities of iodine or polyiodides in both static and dynamic vapor phase as well as aqueous phase condition, which underscore its full potential for large-scale real-time applications. The results revealed that the newly developed hybrid composite is of great scientific significance as a potential adsorbent for effective radioiodine sequestration owing to its rapid uptake kinetics (with rate constant values of $0.0567\,g.g^{-1}h^{-1}$ and $1.9437\,g.mg^{-1}.min^{-1}$ for vapor and aqueous phase iodine sorption, respectively), high sorption capacities ($9.98\,g.g^{-1}$ and $4.74\,g.g^{-1}$ for vapor and aqueous phase iodine sorption, respectively) along with exceptional selectivity, superior retention efficiency (>94% up to 7 days), recovery and satisfying recyclability. We expect that the insight gained in this research are of fundamental importance to understand the rational design and development of innovative hybrid composite materials for efficient sequestration and other applications demanding for porous and macroscopic scaffolds.

## Methods

### Materials

All the reagents, starting materials and solvents were commercially purchased from Sigma-Aldrich, TCI Chemicals, Alfa aesar depending on their availability and used without further purification.

### General characterizations and physical measurements

**Powder X-ray diffraction (PXRD).** Powder X-ray diffraction (PXRD) experiments were performed on a Bruker D8 Advanced X-ray diffractometer at room temperature using Cu Kα radiation ($\lambda = 1.5406\,Å$) at a scan speed of 0.5 ° $min^{-1}$ and a step size of 0.01° in 2θ.

**Fourier transform infrared spectroscopy (FT-IR).** The FT-IR spectra were acquired by using NICOLET 6700 FT-IR spectrophotometer using KBr pellet in 500-4000 $cm^{-1}$ range. The AIR Spectra were acquired by using a Bruker Optics ALPHA-E spectrometer with a universal Zn-Se ATR (attenuated total reflection) accessory. FT-IR/ATR data are reported with a wave number ($cm^{-1}$) scale in 500-4000 $cm^{-1}$ range.

**Thermogravimetric analysis (TGA).** Thermogravimetric analyses were recorded on Perkin-Elmer STA 6000 TGA analyzer by heating the samples from 40 to 800 °C under $N_2$ atmosphere with a heating rate of 10 °C $min^{-1}$.

**Field emission scanning electron microscopy (FESEM).** The morphology of the materials was recorded with a Zeiss Ultra Plus field-emission scanning electron microscope (FESEM) with an integral charge compensator and embedded EsB and AsB detectors (Oxford X-max instruments 80 $mm^2$ (Carl Zeiss NTS, GmbH). The samples were sputter-coated with a 5-10 nm Au film to reduce charging. The elemental analysis was carried out using voltage of 15 KV equipped with an EDX detector. Data acquisition was performed with an accumulation time of >600 s.

**Transmission electron microscopy (TEM).** TEM, High-resolution TEM imaging and STEM-EDS were performed on the HRTEM (JEM-2200FS, JEOL) operating at acceleration voltage of 200 kV. For TEM analysis, all the samples were dispersed in isopropanol (0.5 mg/mL) and sonicated for 15 min. Then, the samples were left for 2 min, and the upper part of the solution was taken for preparing TEM samples on a lacey carbon-coated copper grid (Electron Microscopy Science, 200 mesh).

**Solid-state nuclear magnetic resonance (NMR) spectroscopy.** Solid-state $^{13}C$ cross-polarization-magic angle spinning (CP-MAS) spectra were conducted on a Bruker 500 MHz NMR spectrometer with a CP-MAS probe. Carbon chemical shifts are expressed in parts per million (δ scale).

**Nuclear magnetic resonance (NMR).** $^1H$ NMR spectra were recorded on Bruker 400 MHz NMR spectrometer. The chemical shifts are expressed in parts per million (δ scale).

**High-resolution mass spectroscopy (HRMS).** The mass analysis of the MOPs were carried out using high-resolution mass spectrometry (HRMS-ESI-Q-TOF LC-MS) and Applied Bio system 4800 PLUS matrix-assisted laser desorption/ionization (MALDI) TOF/TOF analyzer.

**Nitrogen adsorption-desorption isotherm measurements.** $N_2$ gas adsorption-desorption measurements were performed using BelSorp-Max instrument (Bel Japan). Prior to adsorption measurements, the activated samples were heated at 120 °C under vacuum for 12 hours using BelPrepvacII.

**UV-visible absorption spectra.** UV-vis absorption studies were performed on a Shimadzu UV 3600 UV /vis /NIR spectrophotometer in an optical quartz cuvette (10 mm path length) over the entire range of 200-800 nm.

**Steady state photoluminescence spectra.** The steady-state photoluminescence studies were recorded on a Fluorolog-3 spectrofluorometer (HORIBA Scientific).

**Fluorescence measurements.** All fluorescence measurements were done on Jobin Yvon Fluoromax-4 spectrofluorometer.

**X-ray photoelectron spectroscopy (XPS).** As-obtained powder samples was stuck to conductive paste and then measured by X-ray photoelectron spectroscopy using K-Alpha+model (Thermo Fischer Scientific, UK) with Al Kα source.

**Zeta potential.** Zeta potential measurements were performed on Anton Paar Litesizer 500 series instrument. Measurement cell: Omega cuvette Mat. No. 155765, Target temperature 25.0 °C, Equilibration time – (Series parameter), Henry factor 1.1 (Other), Adjusted voltage (Automatic Mode), Number of runs 20, Solvent – water.

**Raman measurements.** Raman spectra were acquired with an Xplora PLUS Raman microscope (Horiba Company) (785 nm laser and a 1200 lines/mm grating).

**Raman mapping analysis.** Raman mapping images were acquired with a Renishaw, UK model-Invia Reflex make Raman microscope. Spectrograph equipped with a research-grade microscope capable of producing Raman (wavenumber transfer 50 to 4000 $cm^{-1}$) and PL (330 nm to 1.6 microns). Spectral Range of spectrometer: 200 nm – 1600 nm.

**Supercritical $CO_2$ activation.** Supercritical $CO_2$ activation process was performed using a TOUSIMIS Samdri instrument. A freshly filled liquid $CO_2$ cylinder was used for the experiment.

**Compression test.** The mechanical strength test was conducted with a vernier caliper and each data was collected for three times. Compressive strength was measured in non-confined mode, using universal testing machine (Model Instron 5943) equipped with 1 kN load cell and stainless-steel compression plates. The mechanical compression data was collected at room temperature.

**Electron paramagnetic resonance (EPR) spectroscopy.** (EPR) spectroscopy was carried out on a Bruker EMX plus spectrometer.

**Conductivity measurement.** Four probe DC conductivity measurements were done on pressed pellets of compounds in Keithley 6221 source meter instrument.

**Fluorescence microscopy imaging.** Fluorescence microscopy imaging (Leica Microscopes Model D3) was used to achieve background free high-resolution images of all the compounds. Hamamatsu orca flash4 camera with 20x objective (HC Pl APO 20x/0.80) was used for capturing the digital images of the compound.

**3D X-ray tomography.** Hybrid aerogels were imaged using X-ray microtomography (Xradia 510 Versa X-ray Microscope, Zeiss X-ray Microscopy, Pleasanton, CA, USA) to study its morphology, porosity and pore-size distribution. Specimens were loaded onto the sample holder, kept in between the X-ray source and the detector assembly. Detector assembly consisted of a scintillator, 20X optics and a CCD camera. X-ray source was ramped up to 80 kV and 7 W. The tomographic image acquisitions were completed by acquiring 5001 projections over 360° of rotation with a pixel size of 0.80 microns for a sample size of 0.8 ×0.8 ×0.8 $mm^3$. Each projection was recorded with 3 seconds of exposure time. In addition, projections without the samples in the beam (reference images) were also collected and averaged. The filtered back-projection algorithm was used for the reconstruction of the projections to generate two-dimensional (2D) virtual cross-sections of the specimens. Image de-noising, filtration, segmentation and further processing were performed using GeoDict software package (GeoDict® 2018, Math2Market GmBH, Germany). 2D images were trimmed down to a sub-volume (100 ×100 x 100 voxels with 0.80 cubic micron per voxel), filtered to remove noise and segmented after OTSU threshold selection based on local minima from the grayscale histogram. Resultant 3D reconstructed model was used to estimate the pore characteristics such as porosity, pore-size distribution, etc. using PoroDict® software package (GeoDict® 2018, Math2Market GmBH, Germany), where pore-radius is determined by fitting spheres into the pore volume.

**Synthesis of TPE-NO₂ (tetrakis(4-nitrophenyl)ethylene).** TPE-NO₂ was synthesized by following a previously reported procedure[21]. In an ice bath, 40 mL of acetic acid and 40 mL of fuming nitric acid were added to a 250 mL round bottom flask. Then 5 gm of TPE (15 mmol) was added in small portions over a 20 min period before the reaction was warmed to room temperature with stirring for 3 h. After that the reaction mixture was poured into 300 mL ice water, which yielded yellow precipitation. This precipitate was collected by filtration,

washed with an excess of water, and air dried, affording a light yellow powder with high yield.

**Synthesis of TPE-NH₂(tetrakis(4-aminophenyl)ethylene).** TPE-NH₂ was synthesized by following a previously reported procedure[21]. 2.50 gm of tetrakis(4-nitrophenyl)ethylene (4.34 mmol) was dissolved in 25 mL of anhydrous THF in a 250 mL round bottom flask under nitrogen atmosphere. After that 500 mg of palladium on carbon (wt 10 % Pd) and 25 mL of $NH_2NH_2 \cdot H_2O$ (516 mmol) were added to the solution slowly before the reaction mixture was refluxed for 48 hours. Then the reaction mixture was cooled down to room temperature, and the insoluble residues were filtered off. Further, the solvent of the filtrate was removed under reduced pressure to afford a brown solid of TPE-NH₂ (Supplementary Fig. 1, 2).

**Synthesis of COF aerogel.** The pristine TPE-COF aerogel was synthesized with slight modification based on the report from Rafael Verduzco *et al*[33]. In a reaction glass vial (4 mL), 23.5 mg (0.06 mmol) of 4, 4', 4", 4'''-(ethene-1, 1, 2, 2-tetrayl)tetraaniline [ETTA] and 16.1 mg (0.12 mmol) terephthalaldehyde [TPD] were each dissolved completely in 0.5 mL dimethyl sulfoxide (DMSO), followed by rapid mixing of the solution with the addition of 0.1 mL 6 M acetic acid aqueous solution. Upon addition of acetic acid solution, the mixture became cloudy immediately and underwent a gelation process within a few minutes. Thereafter, the resulting cloudy solution was sealed using Teflon tape to prevent the solvent leakage and further transferred in a closed container and allowed to stand in oven at 80 °C for ~12 h. In this time period the gelation was occurred. The gelation process was further aged for next 12 h at room temperature. After that the wet-gels were removed from the vials and was subjected to wash followed by solvent exchanged with tetrahydrofuran, acetone and ethanol, each for three times. After solvent exchange, to obtain the aerogel, a typical supercritical $CO_2$ (Sc$CO_2$) activation process was performed using a TOUSIMIS Samdri instrument. Thereafter, these aerogels was further reactivated in dioxane (~2 mL) solvent with catalytic amount of 6 M acetic acid (0.1 mL) at 80 °C for another one day. After 24 hours, again, these reactivated aerogels were subjected to solvent exchange and drying under supercritical $CO_2$ condition with the similar protocol mentioned above. Finally, the resulting material was dried under vacuum for overnight to obtain the COF aerogel.

**Synthesis of NH₂-Zr(IV)-MOP.** The NH₂-Zr(IV)-MOP was synthesized with slight modification based on the report from Zhao et al[47]. 2-aminoterephthalic acid (50 mg) and Zirconocene dichloride (150 mg) were dissolved in N, N-dimethylacetamide (DMA, 10 mL) with a trace amount of water (40 drops). This mixture was heated in an oven at 65 °C for 12 hours and then kept at room temperature in undisturbed state for 72 hours. The yellow cubic block crystals were collected by filtration and dried under vacuum.

**General Synthesis of NH₂-Zr(IV)-MOP/COF hybrid composites aerogel (IPcomp-7).** In a typical synthesis procedure, at first, certain amount of -NH₂ functionalized Zr(IV)-MOP was reacted with TPD in ethanolic solution with catalytic amount of 3 M acetic acid for 12 hours at 65 °C under mild-stirring condition. Upon competition of the reaction, the yellow colored precipitate was collected by centrifugation at 4402 x g for 5 min and dried under vacuum. This product was termed as TPD@NH₂-Zr(IV)-MOPs.

The hybrid materials were synthesized by systematic hybridization synthetic strategy via two steps procedure. In the first step, the hybrid wet-gel form of the composite was synthesized. At first, a certain amount of TPD@NH₂-Zr(IV)-MOPs were dissolved in an aqueous dimethyl sulfoxide (DMSO) solution (few drops of $H_2O$ in DMSO). In another vial, precursor of COF, i.e., 23.5 mg (0.06 mmol) of

4, 4′, 4″, 4‴-(ethene-1, 1, 2, 2-tetrayl)tetraaniline [ETTA] and 16.1 mg (0.12 mmol) terephthalaldehyde [TPD] were each dissolved completely in 0.45 mL dimethyl sulfoxide (DMSO). Thereafter, both of these two solution were mixed with each other in a 4 mL reaction glass vial. In this solution, 0.1 mL 6 M acetic acid aqueous solution was introduced, upon which the mixture became cloudy immediately and underwent a gelation process within a few minutes. Thereafter, the resulting cloudy solution was sealed using Teflon tape to prevent the solvent leakage and further transferred in a closed container and allowed to stand in oven at 80 °C for ~12 h. In this time period the gelation was occurred. After, ~12 hours the hybrid wet-gel was formed (Supplementary Fig. 3), which was further aged for another ~12 h at room temperature.

Now, in order to get the hybrid aerogel (IPcomp-7), in the second step, thus-prepared hybrid wet-gel compound was subjected to through solvent exchange with tetrahydrofuran, acetone and ethanol (each exchange for three times) before being carefully transferred into a plastic cell, which was then placed to the supercritical $CO_2$ (ScCO$_2$) activation camber of the automated critical point dryer instrument (TOUSIMIS Samdri). After that, a typical supercritical drying process was performed by addition of ~20 mL of pure ethanol, cooling to 0 to 5 °C and filling the instrument chamber with liquid $CO_2$, followed by purge, soaked in liquid $CO_2$. The purge and soaking cycle (each ~10 min) were repeated for 8 times. Then the temperature of the system was raised to 35-40 °C for next ~2 h. After transforming the system to the supercritical temperature and pressure, it was allowed to slowly bleed overnight. At last, the auto vending of the ScCO$_2$ equipment result the light weight aerogel monolith of the hybrid composite (Supplementary Fig. 3), which was used for further characterization and application. It should be pointed out that the proper number of purge-soaking cycles is essential for the final structure and robustness of the hybrid aerogel monoliths. Thereafter, these hybrid aerogels were further reactivated in dioxane (~2 mL) solvent with catalytic amount of 6 M acetic acid (0.1 mL) at 80 °C for another one day. After 24 hours, again, these reactivated aerogels were subjected to solvent exchange and drying under supercritical $CO_2$ condition with the similar protocol mentioned above. Finally, the resulting material was dried under vacuum for overnight to obtain the hybrid aerogel. In addition to this, different batches of the hybrid composite materials were synthesized by varying the amount of the TPD@NH$_2$-Zr(IV)-MOPs (1 mg, 5 mg, 15 mg, 25 mg, 35 mg, 45 mg, 50 mg, 100 mg) and keeping the fixed amount of the precursors of the COF. Now, for the primary screening test, we performed the vapor phase static iodine capture studies with all of these hybrid materials. As a result, the composite material fabricated with ~15 mg of TPD@NH$_2$-Zr(IV)-MOPs, demonstrated the most efficient capture performance (Supplementary Fig. 38). Therefore, in this work, we chose ~15 mg NH$_2$-Zr(IV)-MOP@COF hybrid aerogel material (termed as IPcomp-7) for further study.

## Data availability
All data used for this study are available in the article and supplementary information files. Additional data are available from the corresponding author upon request.

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

## Acknowledgements

S.F., acknowledges DST-Inspire, W.M., D.M., A.S., acknowledges IISER-Pune and S.L., acknowledges CSIR for research fellowships, respectively. We thank Dr. Arijit Sengupta from the Radiochemistry Division of the Bhabha Atomic Research Center (BARC), India for his assistance with the analysis of the radiation. We thank microscopy facility of IISER Pune for its research facilities. S.K.G thanks DST-SERB project (CRG/2022/001090) for funding.

## Author contributions

S.F. designed the material and performed the synthesis and characterizations. S.F., W.M. and D.M. carried out the initial sorption experiments and analyzed the data. A.T. performed the tomographic experiments. M.M.S. performed the theoretical studies. F.K. performed the confocal studies. S.F wrote the paper, while W.M., D.M., S.L., A.S., and S.K.G. revised the manuscript. All authors discussed the results and commented on the manuscript. S.K.G. supervised the study.

## Competing interests

The authors declare no competing interests.
