## [Peer Review file · Nature Communications]

REVIEWER COMMENTS

Reviewer #1 (Remarks to the Author):

The authors report novel hybrid materials based on covalently linked MOP and COF components. The new materials are hierarchically porous, combining macro, meso, and micropores. The successful isolation of these materials may mark the initial point for the development of a new category of hybrid materials. Thus, the synthetic aspect of this work is particularly important. The authors have conducted extensive characterizations using various state-of-the-art techniques. The graphs and figures are of high quality and visually striking. Additionally, the materials have exhibited an exceptional iodine sorption capacity. The authors have conducted detailed studies, encompassing not only static sorption experiments but also dynamic ones. The results are equally intriguing, highlighting the potential of these materials for various applications. Significantly, the authors have performed comprehensive studies, both theoretical and experimental, to elucidate the mechanism of iodine capture. Overall, the paper is highly innovative and unquestionably deserving of publication in Nat. Commun. I have only a few comments:

1. I suggest that the authors apply a non-linear equation to fit the kinetic sorption data. Linearized models may not be as accurate (please refer to <https://doi.org/10.1016/j.heliyon.2023.e15128> and <https://link.springer.com/article/10.1007/s11783-009-0030-7> for reference). Non-linear fitting can be easily performed with various software packages such as Origin.
2. Typically, a K_d value of 10^4 ml/g and not of 10^3 ml/g (as mentioned by the authors in Line 361, page 15) is considered exceptional.
3. In Fig. 56 and 64 in the Supplementary Information (SI), there are no Langmuir fitting data (these are standard curves, absorbance vs. concentration). The authors should provide Langmuir fitting using non-linear models. As mentioned above, fitting with non-linear equations is preferable.
4. How the authors determine the maximum iodine sorption capacity in seawater? Have they performed studies with variable iodine concentrations and fit the data with an isotherm model?
5. The authors may provide a comparison of the iodine sorption capacity (vapor and aqueous iodine) of the new material vs. those of reported ones. Appropriate tables may be added in supplementary information file.

Reviewer #2 (Remarks to the Author):

The manuscript by Ghosh and co-workers is a thorough study on the design and development of an inorganic-organic hybrid porous composite material for efficient and selective adsorption of iodine species. I agree to the publication of the manuscript in Nature Communications, albeit after the following questions are convincingly addressed.

1. My biggest question is regarding the chemical composition (and the structure) of the hybrid. I am not quite sure of the interaction between the MOP and the COF. The authors suggest that it is a covalently linked hybrid but I do not see how a regular structure (and hence a reproducible crystalline structure) can be formed as suggested in Figure 1. Is it then just the COF framework where the MOP is adsorbed sporadically?
2. Please add a comparison with the simulated COF system. And if it is the COF that is the backbone of the crystalline framework, how do the authors explain the shifting of the stacking reflection at about 20° 2θ to lower angles upon formation of the hybrid?
3. The 200 reflection seems to have been shifted as well. How is this related to the formation of the hybrid?
4. Supplementary Figure 14: Please label the IR signals in accordance with the manuscript text. In the green band, what is the difference between the COF and the hybrid? And how does this IR data imply the formation of a chemically bound 'grafted' composite? How is the green band IR signal telling about the imine bond formation/presence in the composite, as discussed in the text in page 7 of the manuscript?
5. Figure 2c: Are the Zr 3d XPS signals really shifted notably?
6. Supp. Figure 16: At 171 ppm for IPcomp-7, there is hardly any signal above the noise level. If ^{13}C NMR doesn't help, the authors could consider doing a ^{15}N NMR to provide a piece of more convincing evidence.
7. FESEM: The images do not seem to show a fiber-like microstructure. Spongy architecture is okay.

Overall, a good work.

Reviewer #3 (Remarks to the Author):

In this work, the authors design the composite of amine-functionalized MOP with COF aerogel called IPcomp-7 for efficient iodine adsorption. The authors surveyed previous studies on the strategies to enhance the iodine adsorption capacity, which is inspired the design of IPcomp-7 presented in this work. The combination of dense N-containing moieties for iodine enrichment and the cationic nature with free chloride ions for trapping triiodine, as well as the fabrication as aerogel for fast kinetics. The MOP/COF aerogel composite shows high iodine adsorption capacity in both aqueous-phase and vapor iodine with fast adsorption kinetics.

Although the work presented the design of MOP/COF aerogel for enhancing iodine adsorption capacity with fast kinetics and reusability. The iodine adsorption experiments were conducted using both aqueous-phase and vapor iodine. There are several points to be corrected or clarified as follows:

1. Disposing of radioactive iodine isotopes (I-129 and I-131) is undoubtedly vital in several points of view. However, the experiments to evaluate adsorption performances of IPcomp-7 were conducted using non-radioactive iodine isotope. How to ensure that the adsorption performance remains the same for the radioactive isotopes?
2. In some parts of the work, the authors evaluate adsorption capacity by comparing the sample weight before and after iodine adsorption. How to confirm that all of the increased weight is solely from the adsorbed iodine species? If other contaminants occurred during the adsorption test, this would lead to the overestimated adsorption capacity.
3. It is suggested to report more specific data/values from the key findings in the abstract and conclusion of the manuscripts. For example, the best values (i.e. maximum capacity, kinetic rate constant) that reflect the high performance of IPcomp-7 should be stated. The words “high capacity, high retention efficiency, fast adsorption kinetics” are too general and broad.
4. The author should compare/benchmark the performance of IPcomp-7 in terms of maximum capacity, iodine selectivity, and adsorption kinetics with previous reports in the literature. Especially with all references mentioned in this manuscript and those recent publications of MOP/COF materials for iodine adsorption (for example: *Chem. Sci.*, 2021, 12, 8452.; *Chem. Mater.*, 2022, 34, 11062; *Sep. Purif. Technol.*, 2023, 123108).
5. Please provide more discussion to confirm that amine-functionalized MOP is located in the internal pore of COF or the void space of aerogel.
6. Does the amount of vapor iodine vary by the atmosphere or humidity?
7. The author provided only the kinetic rate constants of IPcomp-7 (i.e. supporting Figure 56, 63) without kinetic rate constant values of MOP and COF. It would be difficult to conclude that IPcomp-7 has a faster kinetic than the pristine MOP or COF. The table comparing kinetic rate constants for all samples should be provided.
8. The position of valves 1-3 mentioned in the experimental section of vapor phase dynamic iodine uptake studies should be provided in the photo of the home-built setup shown in supporting figure 40.

9. The dynamic tests with a flow rate of 0.75 mL/min seem too slow for practical use. Please provide the reason for using such a slow flow rate.

10. According to the weight increase of MOP/COF composite after iodine adsorption, is the volume expansion of the composite in the column of dynamic tests observed?

Reviewer-1

General comments: The authors report novel hybrid materials based on covalently linked MOP and COF components. The new materials are hierarchically porous, combining macro, meso, and micropores. The successful isolation of these materials may mark the initial point for the development of a new category of hybrid materials. Thus, the synthetic aspect of this work is particularly important. The authors have conducted extensive characterizations using various state-of-the-art techniques. The graphs and figures are of high quality and visually striking. Additionally, the materials have exhibited an exceptional iodine sorption capacity. The authors have conducted detailed studies, encompassing not only static sorption experiments but also dynamic ones. The results are equally intriguing, highlighting the potential of these materials for various applications. Significantly, the authors have performed comprehensive studies, both theoretical and experimental, to elucidate the mechanism of iodine capture. Overall, the paper is highly innovative and unquestionably deserving of publication in Nat. Commun. I have only a few comments:

Response: We are thankful as well as very appreciative to the esteemed reviewer for taking valuable time to read the manuscript, as well as accepting the manuscript, being persuaded by our findings, being convinced by our conclusions, and ultimately making insightful and constructive remarks. Such encouraging words would undoubtedly propel our future endeavors in this field. All his/her concerns have been carefully addressed in the revised manuscript and we believe that the quality of the revised manuscript has been significantly improved.

Comment 1: I suggest that the authors apply a non-linear equation to fit the kinetic sorption data. Linearized models may not be as accurate (please refer to <https://doi.org/10.1016/j.heliyon.2023.e15128> and <https://link.springer.com/article/10.1007/s11783-009-0030-7> for reference). Non-linear fitting can be easily performed with various software packages such as Origin.

Response 1: We are very grateful to the esteemed reviewer for the aforementioned constructive suggestions. In regard to the comment of the learned reviewer, we have now provided the non-linear equation fitting plots for all the kinetic sorption data in the revised supplementary information file (Supplementary figures 36, 42, 49, 56, 63).

Comment 2: Typically, a K_d value of 10^4 ml/g and not of 10^3 ml/g (as mentioned by the authors in Line 361, page15) is considered exceptional.

Response 2: We are thankful to the learned reviewer for pointing out this mistake. In accordance to the suggestion of the learned reviewer's comment, we have now modified the information of K_d value as ' $K_d > 10^4$ mL.g⁻¹ is considered as exceptional' in the revised manuscript (line 361, page number 15).

Comment 3: In Fig. 56 and 64 in the Supplementary Information (SI), there are no Langmuir fitting data (these are standard curves, absorbance vs. concentration). The authors should provide Langmuir fitting using non-linear models. As mentioned above, fitting with non-linear equations is preferable.

Response 3: We are thankful to the learned reviewer for the aforementioned suggestions. As suggested by the reviewer, we have now provided the Langmuir fitting data plots using non-linear model equation in supplementary figures 57 and 64 of the revised supplementary information file.

Comment 4: How the authors determine the maximum iodine sorption capacity in seawater? Have they performed studies with variable iodine concentrations and fit the data with an isotherm model?

Response 4: In regard to the comment of the learned reviewer, we would like to mention that the main focus of our study is to develop a rationally designed hybrid composite porous material that can be utilized for efficient extraction of molecular iodine or iodide species from vapor phase and solution (both organic solvent and aqueous medium) phase. Therefore, an extensive number of sorption studies in both these phases have been performed. Moreover, owing to the importance of iodine as polyiodide species in water medium, we have also conducted a thorough study (kinetics, capacity, selectivity and others) in pure aqueous phase in detail. In particular, the sorption capacities were calculated from the Langmuir isotherm model fitting curves, which has been performed by taking a range of different concentrations of iodine in solution phase. However, the iodine sorption test in seawater was a proof-of-concept study to demonstrate the efficacy as well as versatility of the material towards its utility in different real water systems. Therefore, the maximum iodine sorption capacity by the composite in seawater was calculated by treating 5 mg of compound with a single saturated concentration of I_2+KI stock solution using seawater. Then, the total

uptake of I_3^- in seawater was calculated from the remaining concentration using UV-vis spectroscopy. Therefore, we would like to humbly submit that in this particular case, the maximum iodine sorption capacity in seawater has not been calculated with variable iodine concentrations and hence was not fitted with isotherm model.

Comment 5: The authors may provide a comparison of the iodine sorption capacity (vapor and aqueous iodine) of the new material vs. those of reported ones. Appropriate tables may be added in supplementary information file.

Answer 5: We are thankful to the learned reviewer for the constructive suggestion. As suggested by the reviewer, we have now provided the table of comparison of iodine sorption capacities (in both vapor phase and solution phase) of various reported state-of-the-art materials with our hybrid composite material in the supplementary information file (Table S7 and S9, page number S41 and S62, respectively).

Reviewer-2

General comment: The manuscript by Ghosh and co-workers is a thorough study on the design and development of an inorganic-organic hybrid porous composite material for efficient and selective adsorption of iodine species. I agree to the publication of the manuscript in Nature Communications, albeit after the following questions are convincingly addressed.

Response: We are very grateful to the esteemed reviewer for appreciating our work as well as accepting the manuscript. All his/her concerns have been carefully addressed and we believe the quality of the revised manuscript has been significantly improved.

Comment 1: My biggest question is regarding the chemical composition (and the structure) of the hybrid. I am not quite sure of the interaction between the MOP and the COF. The authors suggest that it is a covalently linked hybrid but I do not see how a regular structure (and hence a reproducible crystalline structure) can be formed as suggested in Figure 1. Is it then just the COF framework where the MOP is adsorbed sporadically?

Answer 1: We truly appreciate the reviewer's genuine concerns. The learned reviewer comments about the chemical composition or the structure of the hybrid composite material that has been developed in this study. We would like to address this enquiry as follows. In regard to the comment of the learned reviewer, we would like to humbly submit that we tried at our best to thoroughly characterize the hybrid material (IPcomp-7) with all the possible relevant state-of-the-art techniques in order to understand its chemical composition as well as configuration. One of the major challenge in our study was to understand the structure of the hybrid composite that has been developed by reacting the Zr-MOP along with imine functional COF materials. However, after performing a number of thorough characterizations, we successfully established the structure of the hybrid that is constructed via covalent joining of amino functional Zr-MOP with imine-COF as describe below.

At first, the MOP that has been used here for the composite synthesis is highly soluble in aqueous methanolic solution (*J. Am. Chem. Soc.* 2018, 140, 20, 6231–6234). The solution possibility of these MOPs help us in order to understand their stable existence within the COF structure. Briefly, the analysis of the

aqueous supernatant collected from the unwashed pristine composite using ^1H NMR, HRMS, and ICP data revealed the presence of MOP molecules (Supplementary Figure-17). This finding can be attributed to the simple liberation of non-covalently bonded MOP molecules from the surface of IPcomp-7. However, the analysis of IPcomp-7 treated supernatant, showed no presence of MOPs after through washing with aqueous methanolic solution, arguing out the occurrence of excess MOPs on the surface of the composite and the absence of MOP liberation from the COF aerogel matrix (Supplementary Figure-18). Finally, the appearance of Zr metal in the ICP data of digested sample of IPcomp-7 and the relevant peaks of the corresponding organic linkers of MOP in the ^1H NMR spectra demonstrated as a major evidence to confirm the existence of covalently connected amino-Zr(IV)-MOPs in the COF aerogel matrix (Supplementary Figure-19). In addition to this, other characterizations, including FTIR, XPS, ^{13}C CPMAS NMR, N_2 sorption data, etc. also supported the binding of NH_2 -MOP with COF matrix. Furthermore, inspired by the learned reviewer's concern regarding the covalent joining of the MOPs with COF, now we have developed a model compound by reacting the NH_2 -MOP with benzaldehyde (as one of the model monomer of the COF), which support the chemically bound MOP with COF structure formation (Supplementary Section S5.2, Page number S25). Similar observation was reported by Zhang et al. in their previous study (*J. Am. Chem. Soc.* 2019, 141, 12064–12070). Based on all of these investigations, we propose that, in the instance of IPcomp-7, the MOPs are covalently bonded with the COF structure rather than being adsorbed sporadically on the surface of the COF matrix. Additionally, we hypothesized that the formation of the regular structure of the hybrid composite material was similar to that of the pristine crystalline COF system because the crystalline configuration of the COF backbone of IPcomp-7 was not significantly disturbed by the relative small amount of covalently lined nanosized MOPs present in the large COF aerogel structure. Such an observation can be compare with the previously reported metal complex loaded COF systems for various applications. (*J. Am. Chem. Soc.* 2011, 133, 19816–19822), (*Chem* 2020, 6, 3172–3202), (*J. Am. Chem. Soc.* 2017, 139, 47, 17082–17088).

Comment 2: Please add a comparison with the simulated COF system. And if it is the COF that is the backbone of the crystalline framework, how do the authors explain the shifting of the stacking reflection at about 20deg 2theta to lower angles upon formation of the hybrid?

Answer 2: We sincerely thank the esteemed reviewer for bringing this concern to our attention. In response to the learned reviewer's observation, we would like to humbly submit that due to a certain routine change in the instrumental set-up/software the whole PXRD spectra in our prior display of the pristine COF's was found to shift somewhat. However, now, we found no noticeable shifting between the redrawn PXRD data of the pure COF and hybrid composite and the simulated one, after closely comparing them side by side (Supplementary Figure 8). However, in case of stacking reflection (001 facet) at 2theta around 18.7° , was found to slightly shift at lower angles as well as decrease in intensity with compare to the pristine COF system, upon formation of the hybrid material. This might be because of the following reason. The broad peak of (001) suggest the existence of stacking 2D sheets, which were observed with the interlayer distance of the COF (4.8 Å) (AA; eclipsed structure). (*J. Am. Chem. Soc.* 2014, 136, 15885–15888). The formation of the eclipsed structure originates from the strong tendency for the hexagonal/trigonal units of the COF structure to form coplanar aggregates which could stabilize the π - π stacking interactions between the adjacent layers (distance of ~ 4.8 Å). Now, in case of this study, during the formation of the MOP/COF hybrid composite material, few of the covalently linked MOPs might be located or trapped between the adjacent layers of the COF, which further slightly disturb the 2D stacking of the COF layers. Additionally, we believe, the cationic nature of the MOPs in the hybrid composite can also create such shifting of the 2D

layers stacking of the COF backbone. Briefly, when the MOPs are joined with the COF matrix, the interionic repulsion originates from the free excess Cl^- ions of the MOPs can generate increase in the interlayer distance of the stacking of the COF's 2D layers. This further results in the shifting of stacking reflection at 2θ (around 18.7°) to lower angles upon formation of the hybrid. Such observations are common in most of the previously reported COF-based composite systems. (*J. Am. Chem. Soc.* 2011, 133, 19816–19822), (*Chem* 2020, 6, 3172–3202), (*J. Am. Chem. Soc.* 2017, 139, 47, 17082–17088), (*ACS Appl. Mater. Interfaces* 2022, 14, 40, 45669–45678).

Comment 3: The 200 reflection seems to have been shifted as well. How is this related to the formation of the hybrid?

Answer 3: In regard to the comment of the learned reviewer, we would like to humbly submit that during our previous plotting of the PXRD data of the pristine COF, all the peaks (whole spectra) were observed to slightly shift, which was because of change of the instrumental/software set-up. After closely examining the redrawn PXRD data of the pure COF and hybrid composite in comparison to the simulated one, we discovered no discernible shifting between the two, particularly in the case of 200 reflection (Supplementary Figure 8). As a result, its relationship to the composite's formation is negligible.

Comment 4: Supplementary Figure 14: Please label the IR signals in accordance with the manuscript text. In the green band, what is the difference between the COF and the hybrid? And how does this IR data imply the formation of a chemically bound 'grafted' composite? How is the green band IR signal telling about the imine bond formation/presence in the composite, as discussed in the text in page 7 of the manuscript?

Answer 4: We thank the learned reviewer for such suggestion and queries. We want to address this comment as following. At first, the formation of 'covalently' joined or chemically bound NH_2 -MOP with the COF structure has not been claimed based on the FT-IR data only. We report here as "This data (IR) evidenced the grafting of NH_2 -MOP into the COF aerogel matrix along with exposed Zr(IV)-SBU and excessive free amino functional groups.", as discussed in the text in page 7 of the manuscript. However, the chemical bond formation or covalent threading of the amino-MOPs with the TEP-COF framework has been demonstrated by other additional thorough characterizations, such as ^1H NMR, HRMS, ICP spectroscopy data of the digested composite. In addition to the FT-IR, ^{13}C CPMAS NMR and N_2 gas sorption data of the composite, which support the formation of the hybrid material, the ^1H NMR, HRMS and ICP spectroscopy data of the digested composite indicated the covalent joining of the MOPs with COF structure. It should be mentioned, the MOPs are highly soluble in aqueous methanolic solution. The solution possibility of these MOPs help us in order to understand their stable existence within the COF structure. Briefly, the analysis of the aqueous methanolic supernatant collected from the unwashed pristine composite using ^1H NMR, HRMS, and ICP data revealed the presence of MOP molecules (Supplementary Figure-17). This finding can be attributed to the simple liberation of non-covalently bonded MOP molecules from the surface of IPcomp-7. However, the analysis of IPcomp-7 treated supernatant, showed no presence of MOPs after through washing with aqueous methanolic solution, arguing out the occurrence of excess MOPs on the surface of the composite and the absence of MOP liberation from the COF aerogel matrix (Supplementary Figure-18). Additionally, the appearance of Zr metal in the ICP data of digested sample of IPcomp-7 and the relevant peaks of the corresponding organic linkers of MOP in the ^1H NMR spectra confirmed the existence of covalently connected amino-Zr(IV)-MOPs in the hierarchical porous COF aerogel matrix (Supplementary Figure-19). In addition to this, now we have also developed a model compound by reacting the NH_2 -MOP with

benzaldehyde (as one of the model monomer of the COF), which support the chemically bound MOP with COF structure formation (Supplementary Section S5.2, Page number S25). Similar observation was reported by Zhang et al. in their previous study (*J. Am. Chem. Soc.* 2019, 141, 12064–12070). Furthermore, the claim of imine bond formation/presence between the NH₂-MOP and COF aerogel has not been done by FTIR (green band) only in the manuscript. However, the ‘imine bond formation/presence in the composite’ can be discussed based on the IR data, as the pristine COF structure is also made up with excessive imine bonds, one of the major component of the hybrid composite. On the other hand, it should also be point out that distinguishing between the imine bonds of pure COF structure and between NH₂-MOP and COF is difficult from the IR data of the composite, owing to their identical bonding nature. Therefore, in our revised manuscript, we report the presence or grafting of MOPs in the COF structure based on the IR data, not the chemical-imine bond formation between the two components of the composite. Similar observation has been reported in other literature reports (*Adv. Mater.* 2018, 30, 1705454), (*Angew. Chem.* 2018, 130, 12282–12286), (*ACS Cent. Sci.* 2020, 6, 9, 1534–1541). Now, in accordance to the comment of the learned reviewer, we have leveled the relevant IR signals in the supplementary figure 14, which demonstrated the difference between the hybrid composite and pristine components (MOP and COF).

Comment 5: Figure 2c: Are the Zr 3d XPS signals really shifted notably?

Answer 5: We agree with the learned reviewer’s concern regarding the shifting in the peak position of Zr 3d core energy level between the composite and pristine MOP is not significant or notably. Therefore, we have now revised the sentence as “the Zr 3d XPS spectra of the pristine MOP was found to shift slightly after binding with COF-aerogel” in the revised manuscript (page number 7).

Comment 6: Supp. Figure 16: At 171 ppm for IPcomp-7, there is hardly any signal above the noise level. If ¹³C NMR doesn’t help, the authors could consider doing a ¹⁵N NMR to provide a piece of more convincing evidence.

Answer 6: We agree with the learned reviewer’s genuine concern regarding the peak intensity at ~171 ppm in the ¹³C CPMAS NMR spectroscopy data of IPcomp-7. In this regard, we want to humbly submit that after recording the similar data several time, we observed identical peak intensity for the composite, which might be cause of significant less intensity of imine bonds between the components of the hybrid then that of the other peak intensities of the overall composite. Further, we found such similar observation well matches with the other reported composite based literatures (*Adv. Mater.* 2018, 30, 1705454), (*ACS Cent. Sci.* 2020, 6, 9, 1534–1541). Additionally, in order to provide more evidence for the covalent bond formation between the MOPs and COF structure, now we have synthesized a model compound by reacting the amino functional MOP with benzaldehyde (as one of the model monomer of the COF), utilizing similar synthetic protocol (Supplementary Section S5.2, Page number S25). The ¹H NMR data of the model compound, clearly indicated the reaction of NH₂-MOP with aldehyde as the spectra showed the appearance of relevant new peaks in the region of ~7.3-7.7 ppm (Supplementary Figure, Page number S25). Moreover, our observation was found to matches with the previous report by Zhang et al. (*J. Am. Chem. Soc.* 2019, 141, 12064–12070). Therefore, in addition to the unavailability of the facility, we have not consider the ¹⁵N NMR for further analysis in this particular case as the aforementioned ¹H NMR analysis provide adequate evidence.

Comment 7: FESEM: The images do not seem to show a fiber-like microstructure. Spongy architecture is okay.

Answer 7: We are thankful to the reviewer for the suggestion. In accordance to the concern of the learned reviewer, we have now revised the information of morphology of the compound from FESEM image as “sponge-like architecture” instead of “fiber-like microstructure” in the revised manuscript (page number 9).

Overall, a good work.

We express our gratitude and appreciation to the esteemed reviewer for his positive assessment of our work. Such encouraging words would undoubtedly propel our future endeavors in this field.

Reviewer-3

General comment: In this work, the authors design the composite of amine-functionalized MOP with COF aerogel called IPcomp-7 for efficient iodine adsorption. The authors surveyed previous studies on the strategies to enhance the iodine adsorption capacity, which is inspired the design of IPcomp-7 presented in this work. The combination of dense N-containing moieties for iodine enrichment and the cationic nature with free chloride ions for trapping triiodine, as well as the fabrication as aerogel for fast kinetics. The MOP/COF aerogel composite shows high iodine adsorption capacity in both aqueous-phase and vapor iodine with fast adsorption kinetics.

Although the work presented the design of MOP/COF aerogel for enhancing iodine adsorption capacity with fast kinetics and reusability. The iodine adsorption experiments were conducted using both aqueous-phase and vapor iodine. There are several points to be corrected or clarified as follows:

Response: We are very grateful to the esteemed reviewer for taking valuable time to review our work as well as suggesting few points. All his/her concerns have been carefully addressed and the quality of the revised manuscript has been significantly improved.

Comment 1: Disposing of radioactive iodine isotopes (I-129 and I-131) is undoubtedly vital in several points of view. However, the experiments to evaluate adsorption performances of IPcomp-7 were conducted using non-radioactive iodine isotope. How to ensure that the adsorption performance remains the same for the radioactive isotopes?

Answer 1: We are thankful to the reviewer for raising his/her concern. We would like to humbly submit that as the learned reviewer is surely aware of the fact that handling radioactive isotopes require special authorizations and precautions as well as expertise which is not feasible in regular laboratory conditions. As we do not have any easy access to radiation facility we are unable to perform the capture experiments with radioactive iodine isotopes. Additionally, we would also like to mention that in many cases non-radioactive surrogate ions have been utilized in the literature as the suitable substitute to estimate the potential of materials toward radioactive pollutant capture such as non-radioactive perhenate (ReO_4^-) as a surrogate for radioactive pertechnetate (TcO_4^-). (*Nat. Commun.* 2019, 10, 1646), (*Adv. Funct. Mater.* 2022, 2200618), (*Chem* 2020, 6, 2796–2809).

However, in regard to the comment of the learned reviewer, to support the efficiency of the material towards radioactive iodine isotope capture, we have now conducted the similar iodine adsorption test of IPcomp-7

after exposing the adsorbent with different doses of irradiation using 650 kGy/hour. To our delight, we observed that IPcomp-7 was found to retain similar high sorption capacity compared to the non-irradiated pristine sample, which clearly indicate that irradiation has negligible impact on its adsorption capacity (Supplementary Table S8). This experimental finding further support the utility of our composite towards adsorption of iodine in practical radioactive scenario. However, in line with the learned reviewer's speculation, we think maybe under extreme radioactive conditions, there might be a slight drop in the capture performance for IPcomp-7 towards ^{129}I and ^{131}I adsorption, which is out of the scope of this work.

Comment 2: In some parts of the work, the authors evaluate adsorption capacity by comparing the sample weight before and after iodine adsorption. How to confirm that all of the increased weight is solely from the adsorbed iodine species? If other contaminants occurred during the adsorption test, this would lead to the overestimated adsorption capacity.

Answer 2: We truly appreciate the learned reviewer for pointing out this concern. In regard to the comment of the reviewer, we would like to mention that we adopted the standard adsorption procedure for static capture of iodine from vapor phase, which is verified exclusively in most of the previously reported literatures (*Nat. Commun.* 2022, 13, 2878), (*Chem* 2021, 7, 699–714), (*Adv. Mater.* 2018, 30, 1801991), (*Angew. Chem. Int. Ed.* 2021, 60, 22432–22440). In this adsorption procedure, the sorption of iodine by the adsorbent is taken place inside a closed Teflon-made chamber system, in which pure iodine is taken in one glass vial and the adsorbent is taken in another cleaned glass vial. This whole system is then subjected to heating at a certain temperature. The adsorption capacity is calculated by measuring the weight difference before and after heating. Therefore, the possibility of adsorption of contaminants along with iodine from this closed system is negligible, and hence we believe the increase in weight by the material is solely from the adsorbed iodine species only. In addition to this, to cross verify this conclusion, we have now performed a control adsorption test by the material without taking iodine in the similar closed system. As anticipated, we observed no significant change in the weight of the adsorbent (IPcomp-7) after heating (weight before heating = 15.53 mg and weight after heating = 15.47 mg). This result clearly validate the aforementioned conclusion.

Comment 3: It is suggested to report more specific data/values from the key findings in the abstract and conclusion of the manuscripts. For example, the best values (i.e. maximum capacity, kinetic rate constant) that reflect the high performance of IPcomp-7 should be stated. The words “high capacity, high retention efficiency, fast adsorption kinetics” are too general and broad.

Answer 3: We are thankful to the reviewer for the suggestion. In accordance to the concern of the learned reviewer, we have now added the best values, such as maximum capacities and kinetic constant values of IPcomp-7 in the abstract and conclusion of the revised manuscript. We believe that these additions have improved the quality of the manuscript.

Comment 4: The author should compare/benchmark the performance of IPcomp-7 in terms of maximum capacity, iodine selectivity, and adsorption kinetics with previous reports in the literature. Especially with all references mentioned in this manuscript and those recent publications of MOP/COF materials for iodine adsorption (for example: *Chem. Sci.*, 2021, 12, 8452; *Chem. Mater.*, 2022, 34, 11062; *Sep. Purif. Technol.*, 2023, 123108).

Answer 4: We are thankful to the learned reviewer for the suggestion. As suggested by the reviewer, we have now provided the table of comparison of iodine sorption capacities (for both vapor phase and solution phase) of previously reported state-of-the-art materials or benchmark materials with our hybrid composite material (IPcomp-7) in the supplementary information file (Table S7 and S9, page number S41 and S62, respectively). Also, the suggested references have been cited in the revised manuscript with reference number #57, #34, #60.

Comment 5: Please provide more discussion to confirm that amine-functionalized MOP is located in the internal pore of COF or the void space of aerogel.

Answer 5: We are thankful to the learned reviewer for the suggestion. In accordance to the concern of the reviewer, we want to humbly submit that we have thoroughly characterized the hybrid composite material (IPcomp-7) in most of the relevant possible ways. To confirm the stable existence of the amine-functionalized MOPs within the structure / pore of the COF backbone, following characterizations have been performed. At first, FT-IR and ^{13}C CP-MAS-NMR spectroscopy have been conducted to structurally characterized the covalent grafting of the amino functional MOPs with the imine COF matrix (the detail of peak analysis is discussed in the main manuscript, page number 7). In addition to this, now during revision, to further confirm the covalent threading of the MOPs with COF structure, a model compound has been synthesized and characterized (Supplementary Section S5.2, Page number S25). Moreover, ^1H NMR, HRMS and ICP analysis have been performed in order to confirm the presence of MOPs inside the COF structure. On the other hand, the Zr/Cl ratio as $\sim 1:3$ in the EDX data and other optical characterizations of the composite validate the stable existence of the MOPs in the COF structure. Now, the presence of NH_2 -MOPs in the internal pore of COF structure or the void space of COF aerogel has been investigated by performing the low temperature nitrogen (N_2) gas sorption analysis. The significant decrease in the total N_2 gas uptake amount of the composite (pure/washed) with compare to pristine COF aerogel both in the low pressure region (micropores, type-I) and relative high pressure region (mesopores, type-IV) validate that most the MOPs are located in the internal dual pores (micro and meso) of the TPE-COF 2D structures (Figure 2k). Additionally, it should be mentioned that few of the MOPs, which are covalently threaded with the surface of the COF structure can also be located within the void space of the COF aerogel, owing to the macroscopic aerogel nature of the composite. However, as mentioned, most of the MOPs are located in the internal pores of the COF, which is also supported by the decrease in the surface area, pore size distribution and thus pore volume of the composite with respect to pristine COF aerogel (Figure 2l and Supplementary Figure 13). In addition to this, we want to mention that we have proved the presence of covalently joined MOPs in the internal pores of the 2D COF structure by developing another similar-type hybrid composite powder material in one of our recent report (*ACS Cent. Sci.* 2020, 6, 1534–1541). In this case, owing to the typical powder nature of the MOP/COF hybrid composite, most of the MOPs are positioned in the internal pores of the COF, as in this case, there is no possibility of presence of MOPs in the void space of aerogel. Apart from this, few other literatures also support the similar observation (*Chem*, 2023, 9, 1-19), (*Chem* 2020, 6, 2395–2406).

Comment 6: Does the amount of vapor iodine vary by the atmosphere or humidity?

Answer 6: In regard to the comment of the learned reviewer, we would like to mention that we have performed the vapor phase iodine sorption test by IPcomp-7 in both dry and humid conditions (Figure 3b). Under humid condition, we found a small reduction in the adsorption capacities of IPcomp-7 when

compared with dry condition. However, this result demonstrates that humidity had a negligible impact on the ability to efficiently extract iodine from vapor by IPcomp-7.

Comment 7: The author provided only the kinetic rate constants of IPcomp-7 (i.e. supporting Figure 56, 63) without kinetic rate constant values of MOP and COF. It would be difficult to conclude that IPcomp-7 has a faster kinetic than the pristine MOP or COF. The table comparing kinetic rate constants for all samples should be provided.

Answer 7: We are thankful to the reviewer for the suggestion. In accordance to the concern of the learned reviewer, we have now added the table of comparison of kinetic rate constant values of IPcomp-7 along with the pristine materials (MOP and COF aerogel) in the revised supplementary information file (Supplementary Figures 49, 56, 63). These values support the conclusion of relatively faster iodine adsorption rate for IPcomp-7 compared to pristine MOP or COF.

Comment 8: The position of valves 1-3 mentioned in the experimental section of vapor phase dynamic iodine uptake studies should be provided in the photo of the home-built setup shown in supporting figure 40.

Answer 8: We truly appreciate the learned reviewer for pointing out this concern. As per the suggestion of the reviewer, we have now provided the position of valves 1-3 in the schematic diagram of the home-built setup for the vapor phase dynamic mode iodine uptake experiment along with the experimental section of the revised supplementary information file (Supplementary Figure 40).

Comment 9: The dynamic tests with a flow rate of 0.75 mL/min seem too slow for practical use. Please provide the reason for using such a slow flow rate.

Answer 9: We agree with the learned reviewer's concern regarding the flow rate of the column used for the aqueous phase dynamic iodine sequestration study. We would humbly like to submit that the dynamic capture experiment was conducted as a proof-of-concept study to evaluate the practical applicability of IPcomp-7. The flow rate of feed solution (aqueous iodine) for the column test was set as 0.75 mL/min, in order to ensure the maximum contact time of stock solution with the adsorbent, so that complete extraction is achieved. Also, keeping such intermediate flow rate allows sufficient residence time of the analyte solution inside a relatively small column bed allowing satisfactory exposure of the interacting sites present in the composite materials aiding the sorption process. Additionally, it should be mention that such a flow rate for dynamic flow-through based adsorption test, is well reported in the literatures (*Chem. Sci.*, 2016, 7, 2427–2436), (*Chem* 2020, 6, 1–14), (*ACS Appl. Mater. Interfaces* 2022, 14, 20042–20052), (*Chem. Sci.*, 2016, 7, 4804–4824).

Comment 10: According to the weight increase of MOP/COF composite after iodine adsorption, is the volume expansion of the composite in the column of dynamic tests observed?

Answer 10: We thank to the learned reviewer for the comment. In accordance to the concern of the learned reviewer, we want to mention, we do not observed any significant expansion in the volume of the composite/adsorbent, packed in the adsorption cell, when tested for dynamic iodine sorption in vapor phase. However, significant color change from dark yellow to dark black was observed, which is because of adsorption of excess amount of iodine by the adsorbent (Figure S41).

-----End of the report-----

REVIEWERS' COMMENTS

Reviewer #1 (Remarks to the Author):

As I mentioned in my initial review, this is an excellent piece of work, with the authors including detailed characterizations with state of the art techniques as well as high quality graphics. Overall, the authors have addressed my suggestions. The only request I have for the authors is to include the fitting data from the non-linear fitting of kinetics and isotherm and to remove the linear fitting data as those are not accurate enough. In the current manuscript, they have included the graphs with non-linear fitting, but not the non-linear fitting data (i.e. sorption rate constants, maximum sorption capacities etc.). Aside from this, the paper is highly suitable for publication in Nat. Commun.

Reviewer #2 (Remarks to the Author):

The concerns raised have been addressed satisfactorily and the manuscript may now be accepted for publication.

Reviewer #3 (Remarks to the Author):

The authors address all points in the revised manuscript, so I recommend this manuscript for publication.

Reviewer-1

Comment: As I mentioned in my initial review, this is an excellent piece of work, with the authors including detailed characterizations with state of the art techniques as well as high quality graphics. Overall, the authors have addressed my suggestions. The only request I have for the authors is to include the fitting data from the non-linear fitting of kinetics and isotherm and to remove the linear fitting data as those are not accurate enough. In the current manuscript, they have included the graphs with non-linear fitting, but not the non-linear fitting data (i.e. sorption rate constants, maximum sorption capacities etc.). Aside from this, the paper is highly suitable for publication in Nat. Commun.

Response: We are thankful as well as very appreciative to the esteemed reviewer for the aforementioned constructive suggestions. In regard to the comment of the learned reviewer, we have now provided the non-

linear equation fitting data with their corresponding plots for all the kinetic sorption data in the revised manuscript and supplementary information file (Supplementary figures 36, 42, 49, 56, 63).

Reviewer-2

Comment: The concerns raised have been addressed satisfactorily and the manuscript may now be accepted for publication.

Response: We are very grateful to the esteemed reviewer for accepting the manuscript.

Reviewer-3

Comment: The authors address all points in the revised manuscript, so I recommend this manuscript for publication.

Response: We thank the esteemed reviewer for accepting the manuscript.

-----End of the report-----